# SiDyP: Simplex Diffusion with Dynamic Prior for Denoising Llama-Generated Labels

## Abstract

The traditional process of creating labeled datasets is not only labor-intensive but also expensive. Recent breakthroughs in open-source large language models (LLMs), such as Llama-3, have opened a new avenue in generating labeled datasets automatically for various natural language processing (NLP) tasks to provide an alternative to such expensive annotation process. However, the reliability of such auto-generated labels remains a significant concern due to inherent inaccuracies. When learning from such noisy labels, the model's generalization is likely to be harmed as it is prone to overfit those label noises. In this paper, we propose the **Si**mplex Diffusion with a **Dy**namic **P**rior (**SiDyP**) model to calibrate classifier's predication, thus enhancing its robustness towards noisy labels. Our framework leverages simplex diffusion model to iteratively correct noisy labels conditioned on training dynamic trajectories obtained from classifier finetuning. The **P**rior in SiDyP refers to the potential true label candidates which was obtained according to neighborhood label distribution in text embedding space. It is **Dy**namic because we progressively distill these candidates based on the feedback of the diffusion model. Our SiDyP model can increase the performance of the BERT classifier fine-tuned on both zero-shot and few-shot Llama-3 generated noisy label datasets by an average of 5.33% and 7.69% respectively. Our extensive experiments, which explore different LLMs, diverse noise types (real-world and synthetic), ablation studies, and multiple baselines, demonstrate the effectiveness of SiDyP across a range of NLP tasks. We will make code and data publicly (under a CC BY 4.0 license) available on GitHub upon publication of the work.

## 1 Introduction

In the realm of machine learning, the effectiveness of Deep Neural Networks (DNNs) in a variety of applications is largely contingent on the availability of well-annotated datasets (Fisher, 1936; Deng et al., 2009; Touvron et al., 2023a). Traditionally, this annotation process has been carried out manually by subject matter experts (Ratner et al., 2017), ensuring high accuracy but at a substantial cost in terms of time and resources. In response to these constraints, the field has gradually pivoted towards alternative strategies such as active learning (Ren et al., 2021; Kartchner et al., 2020; Yu et al., 2022), transfer learning (Pan & Yang, 2009; Howard & Ruder, 2018), and weak supervision (Stephan et al., 2022; Yu et al., 2020; Lison et al., 2021). These methods help alleviate some of the burdens of manual annotation, yet they often introduce a new challenge: the incorporation of noise in the training data.

The susceptibility of DNNs, especially pre-trained language models to the noise inherent in training data is a formidable challenge, particularly for models like BERT (Devlin et al., 2019b), which can inadvertently fit to inaccuracies. This issue is compounded by weak supervision types—described by Zhou (2018) as incomplete, inexact, and inaccurate supervision—that introduce various forms of label noise. Without appropriate denoising, these models risk learning from erroneous data rather than genuine patterns. Robust denoising strategies, therefore, play a crucial role in refining training datasets. By systematically identifying and amplifying the impact of mislabeled data, these strategies ensure that models are trained on more accurate representations of the data, as demonstrated by efforts in advanced denoising techniques (Ratner et al., 2017; Yu et al., 2020; Zhang et al., 2022; Zhuang et al., 2023).

Transitioning to the era of advanced open-source language models like Llama-3 (Dubey et al., 2024), the capabilities for initial data annotation have seen remarkable improvements (Tan et al., 2024; Yu et al., 2023; Brown et al., 2020). LLMs can generate initial labels for datasets, leveraging its extensive training on diverse textual data. Although numerous methods have been proposed to enhance the capabilities of LLMs, aiming to improve the accuracy and reliability of their annotation (Yu et al., 2023; Yu & Bach, 2023; Wang et al., 2023; Oliveira et al., 2024; Li et al., 2024; Burns et al., 2023), complete immunity to inaccuracies in LLM-generated labels is unattainable, necessitating a robust mechanism to mitigate the harmful impact of their noisy labels. However, LLM-generated label noise is under exploration as previous studies mainly focus on either synthetic noise or real-world noise (Han et al., 2018b; Bae et al., 2022; Zhuang et al., 2023; Wei et al., 2020; Chen et al., 2023a). Synthetic noise is often impractical since it fails to reflect real-world scenarios, where no gold-standard dataset exists for injection. On the other hand, real-world noise is costly to obtain, as it requires subject matter experts (Ratner et al., 2017) to create labeling functions. To bridging this gap, we propose an innovative denoising approach that strengthens classifiers' resilience to LLM-generated noisy labels.

Our approach aims to purify noisy labels via transition matrix-based methods (Patrini et al., 2017; Yao et al., 2021; Zhang et al., 2021b; Xia et al., 2020; Berthon et al., 2021). Adopting the framework from Bae et al. (2022), our denoising method consists of two stages: finetuning pre-trained language classifiers (PLCs) and denoising via generative models. Finetuning a PLC on a noisy dataset yields data's embedding dynamic trajectories (Zhuang et al., 2023) and prior probability $p(\tilde{y}|x)$. By referring to the neighbor's label distribution in embedding space, we are able to collect a list of potential true label candidates and their corresponding weights. We design a simplex diffusion (Mahabadi et al., 2024) label model to reconstruct true labels from noisy labels and training dynamics. The potential true label candidates are refined progressively throughout the training of the diffusion model based on its prediction. The overall framework is presented in Figure 1.

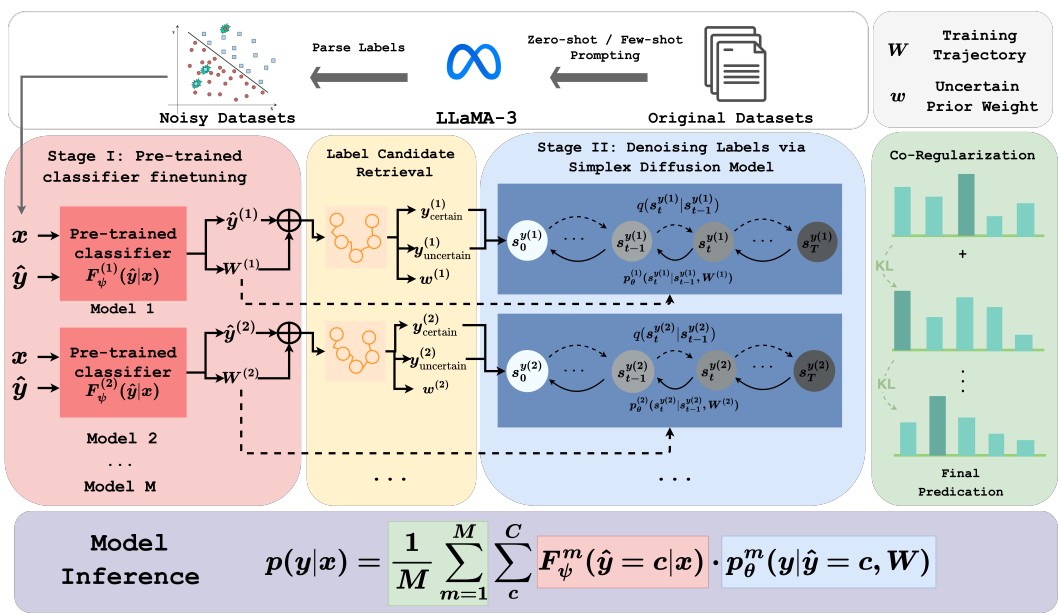

Figure 1: The SiDyP framework, containing (1) pre-trained classifier fine-tuning; (2) dynamic label candidates retrieval and distillation; (3) denoising label using simplex diffusion; (4) co-regularization between multiple model branches; (5) inference process to predict refined labels from noisy labels.

The main contribution of our work include:

- We evaluate previous state-of-the-art baselines, validated on both synthetic and real-world noise, under a novel type of noise: LLM-generated label noise. To the best of our knowledge, this is the first study aimed at enhancing learning under LLM-generated label noise.

- We propose SiDyP, a robust framework using dynamic priors to derive reliable true labels and the simplex denoising label diffusion model to calibrate classifier's predication.

- We conduct extensive experiments of our frameworks compared to 5 state-of-the-art baselines across 4 NLP tasks, 5 LLMs, and 3 different type of noises. Our approach outperforms all the baselines in all the experiments. The effectiveness of each component is also verified.

## 2 BACKGROUND AND MOTIVATION

**Problem Definition**  Let $\mathcal{X} \in \mathbb{R}^d$ and $\mathcal{Y} = \{0, 1, ..., c\}$ be the $d$-dimension input and the target label in a classification task with $c$ classes. Following the joint probability distribution $P$ over $\mathcal{X} \times \mathcal{Y}$, the i.i.d samples forms a gold classification dataset, $\mathcal{D} = \{x_i, y_i\}_{i=1}^N$. Our assumption of learning from noisy labels indicates that the only accessible dataset is $\tilde{\mathcal{D}}_{\text{train}} = \{x_i, \tilde{y}_i\}_{i=1}^N$, sampling from $\tilde{P}$ over $\mathcal{X} \times \tilde{\mathcal{Y}}$ where $\tilde{\mathcal{Y}}$ are potential noisy targets. For a traditional classification problem, the training objective of a classifier $f_\theta$ is to minimize the true risk $R_L(f_\theta) := \mathbb{E}_P[L(f_\theta(x), y)]$. However, in the realm of learning from noisy labels, the only accessible risk function is the noisy empirical risk $\tilde{R}_L^{\text{emp}}(f_\theta) := \mathbb{E}_P[L(f_\theta(x), \tilde{y})]$ due to the absence of true labels $y$. Therefore, our goal is to find a function minimizing the true risk $R_L(f_\theta)$ during learning with noisy empirical risk $\tilde{R}_L^{\text{emp}}(f_\theta)$.

With the only observable target labels being noisy, we manage to train a model that generates probability distribution of true label $y$ given arbitrary input $x$, $p(y|x)$. Taking advantage of noisy labels in our training dataset, we can decompose our objective further as:

$$p(y|x) = \sum_{\tilde{y}} p(\tilde{y}|x)p(y|\tilde{y}, x)$$

In this revised objective, the prior $p(\tilde{y}|x)$ can be directly estimated by finetuning a PLC $\boldsymbol{F_\psi}$ on the accessible noisy dataset. We can approximate the posterior $p(y|\tilde{y}, x)$, expressing the probability distribution of true label $y$ given noisy label $\tilde{y}$ and input $x$, by a generative model. Unlike synthetic noise, which has been extensively studied, LLM-generated label noise is more intricate, contextually influenced, and reflective of real-world class relationships (we include a more detailed discussion in Appendix G). This triggers a more challenging estimation of the posterior as the relation between $\tilde{y}$ and $y$ becomes less predictable and more context-dependent. To tackle this, we begin by focusing on these two key aspects:

1. How can a promising and reliable true label be derived from the noisy dataset?

2. How can we estimate such probabilistic relation between true labels, corrupted labels, and input features accurately?

We define corrputed labels as one which is mislabeled thus incorrect. In the following sections, we introduce our true label candidates dynamic distillation (Section 3) and simplex denoising label diffusion model (Section 4) to address these two concerns respectively. We also adopt training dynamics during PLC fine-tuning and co-regularization mechanism (Appendix C) to make SiDyP tolerant to noises.

## 3 TRUE LABEL CANDIDATES DYNAMIC DISTILLATION

Extracting true labels from a noisy dataset is crucial, as it directly impacts the quality of the subsequent generative posterior approximation. Our derivation of true label is based on the assumption that textual embeddings are robust enough to discriminate between clean and corrupted data samples(Ortego et al., 2021). Texts belonging to the same class typically exhibit similar semantics, making them more likely to cluster together in the embedding space. Therefore, the neighboring labels reveal information about the true labels. Different from prior works(Zhuang et al., 2023; Bae et al., 2022), we retrieve a list of true label candidates for each individual data sample (Algorithm 1). These true label candidates are distilled according to our diffusion model's feedback during training (Algorithm 2).

## 3.1 Label Candidate Retrieval

Our main purpose is re-assigning labels to noisy samples leveraging true label information in embedding space. First, we need to discriminate noisy samples in the dataset. During the PLC fine-tuning in Stage I, there exist training dynamics in embedding space. The noisy samples tend to exhibit larger mean and standard deviation of Euclidean distances towards their assigned labels (incorrect) compared to clean samples (Zhuang et al., 2023). We split the original dataset into $D_{\text{train}}^{\text{noisy}}$ and $D_{\text{train}}^{\text{clean}}$ by cutting off the top $\sigma$ percent of training trajectories, where $\sigma$ is the estimated error rate. We apply K Nearest Neighbor (KNN) algorithm on $D_{\text{train}}^{\text{noisy}}$ with $D_{\text{train}}^{\text{clean}}$ as the reference. Instead of assigning a single deterministic label, a list of label candidates and its corresponding weights (probability) are generated by KNN classifier. We manage to alleviate the uncertainty injected into training of diffusion model in Stage II by two filters: (1) we preserve the candidate if its associated probability greater than a threshold $\lambda$. These data instances are regarded as deterministic instance since their potential true label is single and certain. The remaining data instances are regarded as uncertain and linked with a list of candidates. (2) For uncertain data instances, we extract the two candidates with highest probabilities. If their summation is greater than a specified threshold $\gamma$, we then eliminate other candidates and only preserve these two dominant candidates.

---

**Algorithm 1:** Potential True Label Candidates Retrieval

---

**Input:** $\mathcal{D}_{\text{train}}^{\text{noisy}}$: $\{\mathbf{x_i}, \mathbf{\tilde{y}_i}\}_{\mathbf{i}}^{\mathbf{n}}$, $\mathcal{M}_{\text{train}}, \mathcal{C}_{\text{knn}}, K, \lambda, \gamma$

**Output:** $\mathcal{D}_{\text{train}}^{\text{certain}}$: $\{\mathbf{x_i}, \mathbf{y_i}\}_{\mathbf{i}}^{\mathbf{m}}$, $\mathcal{D}_{\text{train}}^{\text{uncertain}}$: $\{\mathbf{x_i}, (\mathbf{y_i^0}, \mathbf{y_i^1}, \dots)\}_{\mathbf{i}}^{\mathbf{n-m}}$, $\mathcal{W}_{\text{train}}^{\text{uncertain}}$:
$\{(\mathbf{w_i^0}, \mathbf{w_i^1}, \dots)\}_{\mathbf{i}}^{\mathbf{n-m}}$

1 Split $\mathcal{D}_{\text{train}}^{\text{noisy}}$ into $\{\bar{\mathcal{D}}_{\text{train}}^{\text{clean}}, \bar{\mathcal{D}}_{\text{train}}^{\text{noisy}}\}$ according to noisy marker $\mathcal{M}_{\text{train}}$

2 Fit $\bar{\mathcal{D}}_{\text{train}}^{\text{clean}}$ into KNN classifier $\mathcal{C}_{\text{knn}}$

3 Predict $\mathcal{P}_{\text{train}}$ : $\{(\mathbf{p_i^0}, \mathbf{p_i^1}, \dots)\}_{\mathbf{i}}^{\mathbf{n}}$ of entire dataset $\mathcal{D}_{\text{train}}^{\text{noisy}}$ using $\mathcal{C}_{\text{knn}}$ based on $K$ neighbors

4 Initialize $\mathcal{D}_{\text{train}}^{\text{certain}} = \{\}, \mathcal{D}_{\text{train}}^{\text{uncertain}} = \{\}$ and $\mathcal{W}_{\text{train}}^{\text{uncertain}} = \{\}$

5 **for** $i = 0$ *to* $n$ **do**

6 $\quad \mathbf{p_i^{max}} = \max\{(\mathbf{p_i^0}, \mathbf{p_i^1}, \dots)\}$

7 $\quad$ **if** $\mathbf{p_i^{max}} \geq \lambda$ **then**

8 $\quad\quad$ Insert $(\mathbf{x_i}, \mathbf{y_i^{max}})$ into $\mathcal{D}_{\text{train}}^{\text{certain}}$

9 $\quad$ **else**

10 $\quad\quad \mathbf{p_i^{max1}}, \mathbf{p_i^{max2}} = \text{top2}\{(\mathbf{p_i^0}, \mathbf{p_i^1}, \dots)\}$

11 $\quad\quad$ **if** $\mathbf{p_i^{max1}} + \mathbf{p_i^{max2}} \geq \gamma$ **then**

12 $\quad\quad\quad$ Insert $(\mathbf{x_i}, \{\mathbf{y_i^{max1}}, \mathbf{y_i^{max2}}\})$ into $\mathcal{D}_{\text{train}}^{\text{uncertain}}$

13 $\quad\quad\quad \mathbf{p_i^{max1}}, \mathbf{p_i^{max2}} = \text{softmax}(\mathbf{p_i^{max1}}, \mathbf{p_i^{max2}})$

14 $\quad\quad\quad$ Insert $(\mathbf{p_i^{max1}}, \mathbf{p_i^{max2}})$ into $\mathcal{W}_{\text{train}}^{\text{uncertain}}$

15 $\quad\quad$ **else**

16 $\quad\quad\quad$ Insert $(\mathbf{x_i}, \{\mathbf{y_i^0}, \mathbf{y_i^1}, \dots\})$ into $\mathcal{D}_{\text{train}}^{\text{uncertain}}$

17 $\quad\quad\quad$ Insert $(\mathbf{p_i^0}, \mathbf{p_i^1}, \dots)$ into $\mathcal{W}_{\text{train}}^{\text{uncertain}}$

---

## 3.2 Candidate Dynamic Distillation

Our true label candidates distillation is established based on the observation that the generative model gains the capability to calibrate certain amount of noisy data instances after training on our derived deterministic (certain) dataset. Adhere to the observation, we first train our generative model only on deterministic dataset for $\alpha$ warm-up epochs. We rely on such capable model to evaluate our uncertain dataset over a specified iteration $\beta$. During each evaluation, if model's predicted label lies in the candidate lists, the matched label candidate will increase accordingly. The weight list will then be normalized as well to maintain a summation to 1. After candidate weight update and model evaluation for uncertain data samples, we sample a specific label candidate from the candidate list multinomially based on the candidate weights. We treat such a sample label as the true label in this training epoch. The generative model is then trained on both deterministic pair and uncertain pair. Subsequently, the loss of generative model for uncertain sample is weighted by the sampled candidate's weight.

---

**Algorithm 2:** Distill True Label from Candidates during Training

---

**Input:** $\mathcal{G}_{\text{model}}$, $\mathcal{D}_{\text{train}}^{\text{certain}}$: $\{\mathbf{x_i}, \mathbf{y_i}\}_{\mathbf{i}}^{\mathbf{m}}$, $\mathcal{D}_{\text{train}}^{\text{uncertain}}$: $\{\mathbf{x_i}, (\mathbf{y_i^0}, \mathbf{y_i^1}, \dots)\}_{\mathbf{i}}^{\mathbf{n-m}}$, $\mathcal{W}_{\text{train}}^{\text{uncertain}}$:
$\{(\mathbf{w_i^0}, \mathbf{w_i^1}, \dots)\}_{\mathbf{i}}^{\mathbf{n-m}}$, $\alpha$, $E$, $\beta$

**Output:** $\mathcal{G}_{\text{model}}$

1 **for** $e = 0$ *to* $E$ **do**

2     **if** $e \le \alpha$ **then**

3        $\{\bar{\mathbf{y}}_{\mathbf{i}}\}_{\mathbf{i}}^{\mathbf{m}} = \mathcal{G}_{\text{model}}[\{\mathbf{x_i}\}_{\mathbf{i}}^{\mathbf{m}}]$ for $\mathcal{D}_{\text{train}}^{\text{certain}}$

4        loss $= \mathcal{F}_{\text{loss}}[\{\bar{\mathbf{y}}_{\mathbf{i}}\}_{\mathbf{i}}^{\mathbf{m}}, \{\mathbf{y_i}\}_{\mathbf{i}}^{\mathbf{m}}]$

5        Optimize $\mathcal{G}_{\text{model}}$

6     **else**

7        **for** $i = 0$ *to* $\beta$ **do**

8           $\{\bar{\mathbf{y}}_{\mathbf{i}}\}_{\mathbf{i}}^{\mathbf{n-m}} = \mathcal{G}_{\text{model}}[\{\mathbf{x_i}\}_{\mathbf{i}}^{\mathbf{n-m}}]$ for $\mathcal{D}_{\text{train}}^{\text{uncertain}}$

9           **if** $\{\bar{\mathbf{y}}_{\mathbf{i}}\}_{\mathbf{i}}^{\mathbf{n-m}}$ *in* $(\mathbf{y_i^0}, \mathbf{y_i^1}, \dots)$ **then**

10             Increase corresponding $\mathbf{w_i^*}$ by $\frac{1-\mathbf{w_i^*}}{\beta}$

11             $(\mathbf{w_i^0}, \mathbf{w_i^1}, \dots) = \text{softmax}[(\mathbf{w_i^0}, \mathbf{w_i^1}, \dots)]$

12        $\{\mathbf{y_i}\}_{\mathbf{i}}^{\mathbf{n-m}} =$ sample $(\mathbf{y_i^0}, \mathbf{y_i^1}, \dots)$ multinomially according to $\mathcal{W}_{\text{train}}^{\text{uncertain}}$

13        $\{\bar{\mathbf{y}}_{\mathbf{i}}\}_{\mathbf{i}}^{\mathbf{n-m}} = \mathcal{G}_{\text{model}}[\{\mathbf{x_i}\}_{\mathbf{i}}^{\mathbf{n-m}}]$ for $\mathcal{D}_{\text{train}}^{\text{uncertain}}$

14        $\{\bar{\mathbf{y}}_{\mathbf{i}}\}_{\mathbf{i}}^{\mathbf{m}} = \mathcal{G}_{\text{model}}[\{\mathbf{x_i}\}_{\mathbf{i}}^{\mathbf{m}}]$ for $\mathcal{D}_{\text{train}}^{\text{certain}}$

15        certain_loss $= \mathcal{F}_{\text{loss}}[\{\bar{\mathbf{y}}_{\mathbf{i}}\}_{\mathbf{i}}^{\mathbf{m}}, \{\mathbf{y_i}\}_{\mathbf{i}}^{\mathbf{m}}]$

16        uncertain_loss $= \{\bar{\mathbf{w}}_{\mathbf{i}}\}_{\mathbf{i}}^{\mathbf{n-m}} \times \mathcal{F}_{\text{loss}}[\{\bar{\mathbf{y}}_{\mathbf{i}}\}_{\mathbf{i}}^{\mathbf{n-m}}, \{\mathbf{y_i}\}_{\mathbf{i}}^{\mathbf{n-m}}]$

17        loss $=$ certain_loss $+$ uncertain_loss

18        Optimize $\mathcal{G}_{\text{model}}$

---

## 4   SIMPLEX DENOISING LABEL DIFFUSION MODEL

In terms of posterior approximation via generative models, we tackle it from the perspective of denoising diffusion models, which is designed for reconstructing high-fidelity data from pure noise iteratively. We view the true label inference as an progressively denoising process from noisy label based on input feature $x$. In this paper, we apply simplex diffusion model (Mahabadi et al., 2024), one of the continuous diffusion model, to approximate the true label posterior probability from noisy labels. Simplex diffusion model diffuses in simplex probability space, which aligns with our attempt to estimate the posterior distribution.

**Label Simplex Representation**   True label $y$ will be represented in one-hot encoded format $y \in \{0, 1\}^C$. For specific category $c$, $y_c = 1$ and $y_i = 0$ where $i \ne c$. Given the discrete nature of one-hot data representation, we need to first map such categorical data to continuous space to fit our continuous simplex diffusion model. We map the one-hot label representation $y \in \{0, 1\}^C$ to $k$-logit simplex to generate $s^y \in \{\pm k\}^{|C|}$, whose $i$-th component satisfies

$$s_{(i)}^c = \begin{cases} k, & \text{if } i = c, \\ -k & \text{otherwise.} \end{cases} \tag{1}$$

where $k \in \mathbb{R}$ is a hyperparameter.

**Training**   Let $\boldsymbol{y} \in p_{\text{data}}$ be the one-hot representation of a label with $C$ classes and $\boldsymbol{s^y} = \{\pm k\}^{|C|}$ be its $k$-logit simplex representation of $\boldsymbol{y}$. The simplex diffusion model forward process $q(\boldsymbol{s_t^y}|\boldsymbol{s_{t-1}^y})$ is defined as a Gaussian-Markov process that produces a sequence of latent variables $\boldsymbol{s_1^y}, \dots, \boldsymbol{s_T^y}$ by gradually adding Gaussian noise at each time step $t \in 1, 2, \dots, T$ with variance $\beta_t \in \mathbb{R}_{>0}$:

$$q(\boldsymbol{s_t^y}|\boldsymbol{s_{t-1}^y}) = \mathcal{N}(\boldsymbol{s_t^y}|(1-\beta_t)\boldsymbol{s_{t-1}^y}, \beta_t\mathbf{I}) \tag{2}$$

Let $\boldsymbol{\epsilon_t} \sim \mathcal{N}(0, k^2\mathbf{I})$ as we convert data into simplex space, $\alpha_t = 1 - \beta_t$, and $\bar{\alpha}_t = \prod_{j=1}^{t} \alpha_j$. Sampling $\boldsymbol{s_t^y}$ at an arbitrary time step $t$ has a closed-form solution:

$$s_t^y = \sqrt{\bar{\alpha}_t} s_0^y + \sqrt{1 - \bar{\alpha}_t} \epsilon_t \tag{3}$$

Given a well-behaved noise schedule $\{\beta_t\}_{t=1}^T$, a little amount of Gaussian noise with variance $\beta_t$ is injected, while a large amount $1 - \beta_t$ of previous sample $s_{t-1}^y$ is preserved for each time step $t$. At the last time step $t = T$, our original data is expected to be no different from pure Gaussian distribution $\mathcal{N}(0, \mathbf{I})$. Therefore, in the denoising process, we can sample random noise from a standard Gaussian distribution and recover it sequentially to samples from $p_{\text{data}}$. Such an approximation of the reverse process $q(s_{t-1}^y | s_t, s_0)$ can be delivered via a neural network with parameters $\theta$, $p_\theta(s_{t-1}^y | s_t^y)$. In the context of our posterior estimation, neural network is conditioned on $s^{\tilde{y}}$, where $\tilde{y}$ is the noisy label, to approximate $s_{t-1}^y$ at time step $t$. The reverse process then is parameterized as

$$p_\theta(s_{t-1}^y | s_t^y, s^{\tilde{y}}, x) = \mathcal{N}(\mu_\theta(s_t^y, t | s^{\tilde{y}}, x), \Sigma_\theta(s_t^y, t | s^{\tilde{y}}, x)) \tag{4}$$

As cross-entropy loss is typical in classification problem, we adopt it between the ground truth label and the model prediction given a noisy logit simplex $s_t$ at time step $t$.

$$\mathcal{L} = \mathbb{E}_{t, q(s_0^y | s^{\tilde{y}}, x_i), q(s_t^y | s_0^y, s^{\tilde{y}}, x_i)} \left[ - \sum_{i=1}^L \log p_\theta(y_i | s_t^{y_i}, t, s^{\tilde{y}_i}, x_i) \right] \tag{5}$$

**Noise Schedule**  One important component in the diffusion forward process is the noise schedule. We follow the following cosine schedule for $\alpha_t$:

$$\bar{\alpha}_t = \frac{f(t)}{f(0)}, \quad f(t) = \cos\left( \frac{\frac{t}{T} + s}{1 + s} \cdot \frac{\pi}{2} \right)^2 \tag{6}$$

**Inference**  During the inference of the simplex diffusion model, $s_T$ is sampled from the prior $\mathcal{N}(0, k^2 \mathbf{I})$. The model predictions are iteratively denoised for $t = T, \ldots, 1$ starting from $k$-logit simplex Gaussian noise. This reverse process can be approximated via an adjustment of Equation (3):

$$s_{t-1} = \sqrt{\bar{\alpha}_{t-1}} \hat{S}_\theta(s_t, t | s^{\tilde{y}}, x) + \sqrt{1 - \bar{\alpha}_{t-1}} \epsilon_t \tag{7}$$

where $\hat{S}_\theta$ is the model prediction of the ground-truth, $s^{\tilde{y}}$ is noisy label simplex and $x$ is the input embedding, on which the model is conditioned. The model prediction $\hat{S}_\theta(s_t, t | s^{\tilde{y}}, x)$ is regarded as the hypothetical ground-truth and corrupt it by $(t - 1)$ time steps. To construct the model prediction, we project the logits produced by the underlying conditional model via argmax to match the initial $k$-logit representation:

$$\hat{s}_{(i)}^c = \begin{cases} k, & \text{if } i = \text{argmax}(s^y), \\ -k & \text{otherwise.} \end{cases} \tag{8}$$

## 5  EXPERIMENTS & RESULTS

First, we introduce the tasks and datasets (20News Group, NumClaim, TREC, SemEval) that our experiments are conducted on (Section 5.1). Then, we describe our experimental setup (Section 5.2). Subsequently, we present the results of LLMs noise (Section 5.3) and synthetic noise, and real world noise (Section 5.4). Finally, we validate the effectiveness of each component in our framework (Section 5.5).

### 5.1  TASKS AND DATASETS

For our experiments, we include financial numerical claim detection from Shah et al. (2024), question classification from Li & Roth (2002), semantic relation classification task from Hendrickx et al. (2019), and news topic modeling task from Lang (1995). A summary of datasets used with the train-validation-test split is provided in table 1. We provide brief details about each task and dataset in Appendix A.

| Dataset | # Labels | Dataset Size | | |
|---------|----------|-------|-------|------|
| | | Train | Valid | Test |
| NumClaim | 2 | 1715 | 429 | 537 |
| TREC | 6 | 5033 | 500 | 500 |
| SemEval | 9 | 1749 | 178 | 600 |
| 20News | 20 | 9051 | 2263 | 7532 |

Table 1: Summary of datasets used. Dataset size denotes the number of samples in the benchmark.

## 5.2 Experimental Setup

**Baselines** We compare SiDyP with the most relevant state-of-the-art baselines from three different categories in the realm of learning from noisy labels: (1) *Basic Performances* without specific design tackling noisy labels (Devlin et al., 2019a); (2) *Multi-Model Training Strategies*: **Co-Teaching** (Han et al., 2018a) and **JoCoR** (Wei et al., 2020). **Co-Teaching** trains two networks simultaneously and selects small-loss instances as clean samples for subsquent training. **JoCoR** also trains two networks simultaneously and use co-regularization to achieve agreement to filter out noisy samples by selecting instances with small losses; (3) *Generative Models for Noisy Maxtrix Estimation*: **NPC** (Bae et al., 2022) and **DyGen** (Zhuang et al., 2023). **NPC** utilize a generative model to calibrate the prediction of classifiers trained on noisy labels via a transition matrix. **DyGen** leverages the training dynamics to detect noisy samples and use a generative model to calibrate.

**Evaluation** We evaluate all the experiments using accuracy on clean test datasets. We only run the model on the test dataset at the point when the validation accuracy achieves the highest during training. The reported test performances of all baselines and our SiDyP is selected by this procedure. Given that the success of existing weakly-supervised learning methods relies heavily on clean validation samples (Zhu et al., 2023), we use noisy validation sets for model selections in all experiments. All experiments are run under 5 random seeds. We report the mean of the performances and the standard deviation.

**Implementation Details** We implement SiDyP using PyTorch (Paszke et al., 2019) and HuggingFace (Wolf et al., 2020). We use BERT (Devlin et al., 2019a) as our PLC in Stage I. For our baselines which contains PLC fine-tuning on noisy label datasets (**NPC**, **DyGen**, **GaDyP**), we use only one coherent PLC results for their individual post process to ensure a fair comparison as random seeds affect network initialization, synthetic noise generation, etc. More training details are revealed in Appendix D.

## 5.3 LLMs Noise Experiments

We run extensive experiments on various tasks and diversified LLM noises. First, we examine our framework in NumClaim, TREC, and SemEval labelled by `Llama-3-70b-chat-hf` (Dubey et al., 2024) in both zero-shot and few-shot manner. We only prompt 20News Group in zero-shot manner as it is a document level task, and Llama-3-70b has a context length limitation of 8192, which is not sufficient for few-shot learning. Then, to test SiDyP under diversified LLM noises, we prompt `Meta-Llama-3.1-70B-Instruct-Turbo` (Dubey et al., 2024), `Meta-Llama-3.1-405B-Instruct-Turbo` (Dubey et al., 2024), `gpt-4o` (OpenAI et al., 2024), and `Mixtral-8x22B-Instruct-v0.1` (Jiang et al., 2024) in both zero-shot and few-shot prompting manners on SemEval task. We address the experiment details and results in the following.

**LLM Prompting** For both zero-shot and few-shot manners, we use same prompts of same tasks for different LLMs (See prompting details in Appendix B.2). Notably, when prompting the LLM to label data, it is not guaranteed that it would follow the instructions and output in the specified format. It leads to missing labels for some data samples in our annotated datasets. Although we observe that the portion of missing labels is trivial (i.e. highest missing label ratio (only 0.014%) happens in 20News Group dataset. See full statistics in Appendix E), we still want to preserve those data samples to maintain data's integrity for training. Therefore, we randomly assign a label to those

missing-label samples according to a uniform distribution over all labels. We use the dataset after random assignment for both training and validation. We do not apply random assignment for test dataset and report LLMs' raw accuracy in Table 2 and 3.

**Results**   Table 2 shows the results of Llama-3-70b on all four tasks. Our method (SiDyP) outperforms all baselines by a notable margin 2.05& across all tasks in both prompting manners. There are averagely 6.34% samples of a fine-tuned PLC, and 5.77% of raw Llama-3-70b labelled samples successfully corrected by SiDyP. The performance gain on SemEval task is the most significant, achieving an average increase of 3.7%. This indicates that SiDyP is robust to high noise ratio dataset. Although the base performance of NumClaim is competitive, SiDyP is able to bring an average of 20.19% marginal increase. For NumClaim in few-shot manner, our method is the only one to outperform Llama-3-70b raw labelling accuracy and fine-tuned PLC. We also observe that both methods of multi-model training strategies struggle in these tasks. We think it's because of its training from scratch as PLC possesses prior knowledge that would be helpful despite that they are prone to noisy labels. Transition matrix-based methods performs generally better as it leverages pre-trained models and calibrate it via a post-process.

| Datasets ($\rightarrow$) | NumClaim | | TREC | | SemEval | | 20News |
|---|---|---|---|---|---|---|---|
| Method ($\downarrow$) | Zero-shot | Few-shot | Zero-shot | Few-shot | Zero-shot | Few-shot | Zero-shot |
| Llama-3-70b | 89.94 | 95.53 | 81.80 | 84.00 | 47.50 | 48.50 | 74.04 |
| PLC | 90.54$\pm$0.72 | 95.11$\pm$0.30 | 80.64$\pm$0.94 | 77.72$\pm$1.34 | 51.59$\pm$0.44 | 50.46$\pm$0.72 | 71.2$\pm$0.52 |
| Co-teaching | 82.31$\pm$1.11 | 83.77$\pm$4.05 | 69.20$\pm$2.09 | 67.20$\pm$2.21 | 46.53$\pm$4.16 | 44.29$\pm$6.18 | 35.28$\pm$12.18 |
| JoCoR | 83.35$\pm$1.97 | 85.82$\pm$2.05 | 70.80$\pm$3.00 | 65.82$\pm$2.17 | 45.66$\pm$3.25 | 44.11$\pm$2.23 | 42.39$\pm$11.98 |
| NPC | 90.83$\pm$0.62 | 95.04$\pm$0.61 | 79.48$\pm$1.97 | 78.88$\pm$1.47 | 50.73$\pm$1.70 | 47.53$\pm$1.26 | 70.60$\pm$0.51 |
| DyGen | 91.13$\pm$0.30 | 95.41$\pm$0.28 | 82.88$\pm$0.71 | 84.80$\pm$0.86 | 60.86$\pm$0.81 | 60.79$\pm$2.23 | 71.42$\pm$0.31 |
| **SiDyP** | **93.63**$\pm$**0.84** | **95.97**$\pm$**0.15** | **84.76**$\pm$**0.79** | **85.60**$\pm$**0.44** | **64.26**$\pm$**0.27** | **64.79**$\pm$**0.96** | **72.66**$\pm$**0.58** |

Table 2: Performance comparison of Llama-3-70b on zero-shot and few-shot learning tasks across multiple datasets, including NumClaim, TREC, SemEval, and 20News. Results are reported as classification accuracy with mean and standard deviations of 5 runs under different seed. **Bold** represents the best performance, while underline presents the second-best performance. Same seed setting and presentation apply in the following tables.

**Robustness Check for Diversified LLMs**   Instead of limiting to Llama-3-70b, we extend our experiments to a variety of LLMs of different families with different sizes. We follow the same prompting and assignment procedure as describe above (See details in Appendix B.1). We aim to check the robustness of our SiDyP framework under multiple LLM-generated label noise. Table 3 shows the results of various types of LLM label noise on SemEval. Our method (SiDyP) achieves a significantly better performance compared to all baselines across all LLMs and both prompting manners. Specifically, SiDyP obtain an average of 4.47% performance gain than the second best baseline. Comparing to a fine-tuned PLC on noisy dataset, our method is able to boost the performance by an average of 8.02%. Notably, a significant average increase of 11.73% than LLMs raw accuracy is brought by our method. Combining all, we validate that our method is robust and resilient to different types of LLM noise and different prompting methods.

5.4   SYNTHETIC AND REAL-WORLD NOISE EXPERIMENTS

Observing significant performance improvement in LLM-generated label noises, we further test our method under different families of noises, synthetic and real-world, on SemEval task. We reveal the experiment details and results below.

**Noise Generation**   We inject three types of synthetic noises, including **Symmetric Noise (SN)**, **Asymmetric Noise (ASN)**, and **Instance-Dependent Noise (IDN)**. Symmetric Noise flips labels

| Dataset ($\rightarrow$) | SemEval | | | | | | | |
|---|---|---|---|---|---|---|---|---|
| | Llama-3.1-70b | | Llama-3.1-405b | | GPT4o | | Mixtral-8x22b | |
| Method ($\downarrow$) | Zero-shot | Few-shot | Zero-shot | Few-shot | Zero-shot | Few-shot | Zero-shot | Few-shot |
| Base | 52.66 | 55.16 | 55.16 | 52.16 | 56.50 | 57.66 | 42.66 | 40.83 |
| PLC | 60.26±0.89 | 57.70±1.10 | 54.76±1.24 | 53.96±0.12 | 58.63±0.86 | 61.56±0.93 | 49.29±1.31 | 46.33±1.32 |
| Co-teaching | 52.50±5.35 | 54.09±3.56 | 45.51±1.96 | 51.36±0.89 | 52.13±5.36 | 60.91±5.58 | 39.3±6.79 | 27.35±2.55 |
| JoCoR | 45.06±0.97 | 44.26±9.55 | 45.39±4.29 | 50.28±3.07 | 53.31±5.43 | 53.05±4.78 | 32.94±8.73 | 27.26±1.46 |
| NPC | 60.13±0.77 | 57.49±3.00 | 55.06±2.99 | 54.53±1.24 | 59.56±0.90 | 61.40±1.53 | 47.56±1.26 | 41.96±0.70 |
| DyGen | 68.53±0.88 | 64.53±2.85 | 59.69±1.31 | 51.69±2.02 | 62.63±0.91 | 64.03±0.82 | 50.63±6.43 | 40.23±1.41 |
| **SiDyP** | **71.66±0.91** | **67.43±1.36** | **62.76±0.99** | **60.46±2.06** | **66.86±0.48** | **68.83±1.07** | **57.96±1.94** | **50.66±2.02** |

Table 3: Performance comparison of Llama-3.1-70b, Llama-3.1-405b, GPT4o, and Mixtral-8×22b on zero-shot and few-shot learning tasks on SemEval. "Base" represents LLM's raw accuracy on test sets.

uniformly to other classes (Zhuang et al., 2023; Bae et al., 2022; Han et al., 2018a). Asymmetric Noise flips labels with similar classes (Zhuang et al., 2023; Bae et al., 2022). Instance-Dependent Noise flips label with a probability proportional to the features of the sample (Zhuang et al., 2023; Bae et al., 2022). As synthetic noise is controlled, we use the noise ratio of 50% to make a comparison with LLM noise. We choose 50% because LLM noises ratio on SemEval are around 50%. For real-world noise, we take majority vote on the 164 labeling functions' output provided in WRENCH (Zhang et al., 2021a) for the SemEval dataset.

**Results** In Table 4, we present the results of various synthetic noises and real-world noises on SemEval. SiDyP achieves an average of 2.80% increase compared to the second-best baseline. We observe that the performance increase between SiDyP and a strong baseline DyGen on LLM noises (5.21%) is higher than it on synthetic noises (3.26%). This is because DyGen performs better on synthetic datasets as such noises are less intricate (Zhuang et al., 2023). It further validates that LLM-generated label noises align more with real-world noise, making it more challenging for

| Datasets ($\rightarrow$) | SemEval | | | |
|---|---|---|---|---|
| Method ($\downarrow$) | SN | ASN | IDN | Real World |
| Base | 50.00 | 50.00 | 50.00 | 82.50 |
| PLC | 65.06±2.13 | 40.96±2.60 | 59.83±2.65 | 84.13±0.68 |
| Co-teaching | 49.78±7.82 | 38.79±9.04 | 37.00±3.88 | 70.2±0.7 |
| JoCoR | 51.66±7.88 | 44.84±4.75 | 41.91±6.64 | 69.71±1.17 |
| NPC | 57.73±3.61 | 42.60±5.46 | 54.16±4.91 | 81.23±1.88 |
| DyGen | 73.06±2.07 | 53.16±5.46 | 71.40±1.80 | 82.3±0.13 |
| **SiDyP** | **74.26±1.99** | **59.63±3.06** | **73.19±2.22** | **85.86±0.52** |

Table 4: Performance comparison on SemEval with synthetic noise (SN, ASN, IDN) and real-world noise.

other baselines to arrive at accurate estimates. SiDyP, on the other hand, is resilient to all types of label noise, and brings improvement consistently. Moreover, all baselines are prone to the real-world noise as they struggle to be comparable with Base and PLC performances. SiDyP is the only one outperforming them by 3.36% and 1.73% increase respectively.

## 5.5 Effectiveness of Different Components

We investigate the effectiveness of each component in our SiDyP framework on Llama-3-70b labelled SemEval dataset in both zero-shot and few-shot manners. We eliminate them individually to validate their impact on performances: (1) Replacing our dynamic distillation priors with fix certain priors (for each sample, it's only associated with one fix certain label) in Stage II; (2) Substituting Stage II's generative model, simplex diffusion model with Dirichlet variational auto-encoder (VAE) (Joo

et al., 2019) and Gaussian diffusion model (Sohl-Dickstein et al., 2015; Han et al., 2022; Chen et al., 2023b). Table 5 indicates the result. All experiments are conduct using same PLC fine-tuned results, and share the same value of hyper-parameters. Our simplex denoising label diffusion model surpasses Dirchlete VAE by an average of 2.17%. We believe such an enhancement comes from the de-noising capability of diffusion model. Moreover, it outperforms the Gaussin diffusion model by 8.58%. Our simplex denoising label diffusion model, which diffuses in probability simplex space, constructs a more reliable and accurate label probability from noisy labels. Besides, our dynamic prior distillation brings 1.53% increase. We further validate the improvement source of our dynamic prior by comparing the portion of correct labels we collect with fix prior method (See Appendix F for more details). Combining all, it confirms that our candidate retrieval algorithm could derive more true labels, and our prior distillation could find the correct labels among the candidates.

## 6 RELATED WORK

Weak-supervision in machine learning includes incomplete, inexact, and inaccurate categories, each tailored to specific imperfections in data (Zhou, 2018). Inexact supervision deals with broad labels, while inaccurate supervision, where labels are erroneous, employ techniques like data programming (Ratner et al., 2017), human-in-the-loop strategies (Zhang et al., 2022), and contrastive loss for enhanced learning from data similarities and differences (Yu et al., 2020). Zhang et al. (2021a) apply a two-stage model to manage inaccurate supervision, initially denoising data before training on refined labels. In the landscape of learning from noisy labels, Iscen et al. (2022) proposed that there supposed to be similarities among training instances in the feature/embedding space, leading to the

| Datasets ($\rightarrow$) | SemEval | |
|---|---|---|
| Method ($\downarrow$) | Zero-shot | Few-shot |
| FP + Dir-VAE | $60.86_{\pm 0.81}$ | $60.79_{\pm 2.23}$ |
| FP + Sim-Diff | $\underline{62.73_{\pm 1.06}}$ | $\underline{63.26_{\pm 1.06}}$ |
| DP + Gau-Diff | $54.53_{\pm 3.48}$ | $57.36_{\pm 3.64}$ |
| DP + Sim-Diff (**SiDyP**) | $\mathbf{64.26_{\pm 0.27}}$ | $\mathbf{64.79_{\pm 0.96}}$ |

Table 5: Different components efficacy on zero-shot and few-shot labelled SemEval by Llama-3-70b. "FP"=fix prior. "DP"=our dynamic prior. "Dir-VAE"=Dirchlete VAE. "Gau-Diff"=Gaussian diffusion model. "Sim-Diff"=simplex diffusion model.

consistency of labels between data instances and their neighbors. NPC proposed by Bae et al. (2022), lies in the class of transition matrix base method. The true label is inferred by a prior, estimated by a pretrianed classifer, and a posterior, approximated by a generative model. DyGen (Zhuang et al. (2023)) infers true label based on the training dynamics during finetuning the pretrained language model. The feasibility of Diffusion Models in classification problems are explored and validated by Han et al. (2022). Chen et al. (2023a) is the very first to exploit the Gaussian diffusion model in the context of noisy label learning. LLMs have also been leveraged to iteratively expand label space under extremely weak supervision. X-MLClass (Li et al., 2024) demonstrated significant improvements in label discovery and multi-label classification accuracy in open-world settings. Additionally, explanation-aware ensembling methods like EASE (Yu et al., 2023) further illustrate how LLMs can be used to improve in-context learning by effectively guiding predictions and mitigating label noise.

## 7 DISCUSSION

In this paper, we propose a denoising framework, SiDyP, to enhance the learning from Llama-3 generated labels noise. Leveraging the principle of partial label learning and neighbor consistency, our label candidate retrieval and prior dynamic refinement algorithm alleviate the harm of incorrect labels during the training of a classifier. We introduce a simplex diffusion model to reconstruct categorical label data and utilize it as a posterior probability distribution estimator to calibrate the inaccurate prior distribution. Our framework boosts few-shot Llama-3 classification accuracy by a 7.69% average increase across all datasets of diverse noise ratios. We believe that our work sheds light on the realm of employing the diffusion model in the context of learning from noisy labels as well as the topics of calibrating incorrect llm-generated datasets.

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

## A    Dataset and Task Detail

- **Numerical Claim Detection (NumClaim)**: This involves extracting numerical claims from financial texts like analysts' reports to forecast stock price volatility. Using a dataset with binary labels for sentences, this task distinguishes between "in-claim" sentences that predict financial outcomes and "out-of-claim" sentences that state factual information.

- **Question Classification (TREC)**: This task involves classifying questions into predefined categories based on their intent and content, as outlined in the TREC dataset from Li & Roth (2002) study. Using a dataset of labeled questions, this task assigns each question to one of six categories: location, entity, description, human, numeric value, and abbreviation. The goal is to determine the type of answer each question seeks, thereby facilitating targeted information retrieval and enhancing the efficiency of question-answering systems.

- **Semantic Relation Extraction (SemEval)**: This task focuses on the multi-way classification of semantic relations between pairs of nominals, as defined in SemEval-2010 Task 8 (Hendrickx et al., 2019). Utilizing a dataset where each pair of nominals is annotated with one of nine (Cause-Effect, Instrument-Agency, etc.) possible semantic relations, this task involves determining the specific type of relationship that exists between the two terms. The nine categories include Cause-Effect, Instrument-Agency, Product-Producer, Content-Container, Entity-Origin, Entity-Destination, Component-Whole, Member-Collection, and Message-Topic. The objective is to enhance the understanding of linguistic patterns and to improve the semantic analysis capabilities of natural language processing systems.

- **News Topic Modeling (20News)**: This task involves classifying news articles into different topics using the well-known 20 Newsgroups dataset (Lang, 1995). The dataset contains around 20,000 documents collected from newsgroups, organized into 20 different categories such as 'rec.sport.baseball', 'comp.graphics', and 'sci.med'. Each document is assigned to one of these categories. The task's objective is to train models to effectively capture the topical structure of news articles, which helps improve text categorization and topic detection capabilities in natural language processing applications.

## B    LLM Prompting Details

### B.1    Model Implementation Details

We use the `Llama-3-70b-chat-hf` (Touvron et al., 2023b) model for all of our inferences. We take advantage of API from together.ai. We are grateful to them for providing free credits and making it possible. We use the model with a $temperature$ value of 0.00 (for reproducibility) and $max\_token$ of 100. The same hyper-parameters are used for `Meta-Llama-3.1-70B-Instruct-Turbo`, `Meta-Llama-3.1-405B-Instruct-Turbo`, `Mixtral-8x22B-Instruct-v0.1`, and `gpt-4o`.

### B.2    Prompt Templates

**Numerical Claim Detection**
    We use the following zero-shot prompt for numerical claim detection:

prompt_json = [

"role": "user", "content": f"Classify the following sentence into 'INCLAIM', or 'OUTOFCLAIM' class. 'INCLAIM' refers to predictions or expectations about financial outcomes, it can be thought of as 'financial forecasts'. 'OUTOFCLAIM' refers to sentences that provide numerical information or established facts about past financial events. Now, for the following sentence provide the label in the first line and provide a short explanation in the second line. The sentence: sentence",

]

We use the following few-shot prompt for numerical claim detection:

prompt_json = [

"role": "user", "content": f"Classify the following sentence into 'INCLAIM', or 'OUTOFCLAIM' class. 'INCLAIM' refers to predictions or expectations about financial outcomes, it can be thought of

as 'financial forecasts'. 'OUTOFCLAIM' refers to sentences that provide numerical information or established facts about past financial events. Here are two examples: \nExample 1: consolidated total capital was $2.9 billion for the quarter. // OUTOFCLAIM\nExample 2: we expect revenue growth to be in the range of 5.5% to 6.5% year on year. // INCLAIM \nNow, for the following sentence provide the label in the first line and provide a short explanation in the second line. The sentence: {sentence}",

]

**TREC**

We use the following zero-shot prompt for the TREC dataset:

prompt_json = [

"role": "user", "content": f"For the following question, which belongs to a specific category, categorize it into one of the following classes based on the type of answer it requires: Abbreviation (ABBR), Entity (ENTY), Description (DESC), Human (HUM), Location (LOC), Numeric (NUM). Provide the label in the first line and provide a short explanation in the second line. The question: {question},

]

We use the following few-shot prompt for the TREC dataset:

prompt_json = [

"role": "user", "content": f"For the following question, which belongs to a specific category, categorize it into one of the following classes based on the type of answer it requires: Abbreviation (ABBR), Entity (ENTY), Description (DESC), Human (HUM), Location (LOC), Numeric (NUM). Here are six examples:\nExample 1: how did serfdom develop in and then leave russia ? // DESC\nExample 2: what films featured the character popeye doyle ? // ENTY\nExample 3: what contemptible scoundrel stole the cork from my lunch ? // HUM\nExample 4: what is the full form of .com ? // ABBR\nExample 5: what sprawling u.s. state boasts the most airports ? // LOC\nExample 6: when was ozzy osbourne born ? // NUM \nNow for the following question provide the label in the first line and provide a short explanation in the second line. The question: {question},

]

**SemEval**

We use the following zero-shot prompt for the SemEval dataset:

prompt_json = [

"role": "user", "content": f"The task is to identify the type of semantic relationship between two nominals in a given sentence. Below are the definitions of the nine relationship categories you must choose from:\nCause-Effect (CE): An event or object leads to an effect.\nInstrument-Agency (IA): An agent uses an instrument.\nProduct-Producer (PP): A producer causes a product to exist.\nContent-Container (CC): An object is physically stored in a delineated area of space.\nEntity-Origin (EO): An entity is coming or is derived from an origin (e.g., position or material).\nEntity-Destination (ED): An entity is moving towards a destination.\nComponent-Whole (CW): An object is a component of a larger whole.\nMember-Collection (MC): A member forms a nonfunctional part of a collection.\nMessage-Topic (MT): A message, written or spoken, is about a topic.\nFor the provided sentence below, determine the most accurate relationship category based on the descriptions provided. Respond by selecting the label (e.g., CE, IA, PP, etc.) that best matches the relationship expressed in the sentence. Provide the label in the first line and provide a short explanation in the second line. The sentence: {sentence},

]

We use the following few-shot prompt for the SemEval dataset:

prompt_json = [

"role": "user", "content": f"The task is to identify the type of semantic relationship between two nominals in a given sentence. Below are the definitions of the nine relationship categories you must choose from:\nCause-Effect (CE): An event or object leads to an effect. (Example: As the right front

wheel of Senna 's car hit the wall , the violent impact caused a torsion on the steering column , causing it to break .)\nInstrument-Agency (IA): An agent uses an instrument. (Example: The necromancer wields the power of death itself , a power no enemy can stand against .)\nProduct-Producer (PP): A producer causes a product to exist. (Example: This website , www.fertilityuk.org , shows how to interpret the changes that take place in the mucus secretions produced by the cells lining the cervix .)\nContent-Container (CC): An object is physically stored in a delineated area of space. (Example: I sent you a suitcase with cash in it so you can fill it up with wine gummies .)\nEntity-Origin (EO): An entity is coming or is derived from an origin (e.g., position or material) (Example: I have always felt so relieved that Roy and the boys had left the creek .).\nEntity-Destination (ED): An entity is moving towards a destination. (Example: The machine blows water into the connecting conduit .)\nComponent-Whole (CW): An object is a component of a larger whole. (Example: He noticed a speck of blood on the man 's thumb and what he thought were several corresponding drops on the driver 's door of the truck .)\nMember-Collection (MC): A member forms a nonfunctional part of a collection. (Example: With the conquest of Jerusalem in 1099 , Geoffrey de Bouillon established a chapter of secular canons in the basilica of the Holy Sepulcher to offer the sacred liturgy according to the Latin rite .)\nMessage-Topic (MT): A message, written or spoken, is about a topic. (Example: A number of scientific criticisms of Duesberg 's hypothesis were summarised in a review article in the journal Science in 1994 .)\nFor the provided sentence below, determine the most accurate relationship category based on the descriptions provided. Respond by selecting the label (e.g., CE, IA, PP, etc.) that best matches the relationship expressed in the sentence. Provide the label in the first line and provide a short explanation in the second line. The sentence: {sentence},

]

**20News**

We use the following zero-shot prompt for the 20News dataset:

prompt_json = [

"role": "user", "content": f"The task is to classify the given text into one of the 20 news group categories. Below are the 20 categories you must choose from:\n1. 'alt.atheism': Discussions related to atheism.\n2. 'comp.graphics': Topics about computer graphics, including software and hardware.\n3. 'comp.os.ms-windows.misc': Discussions about the Microsoft Windows operating system.\n4. 'comp.sys.ibm.pc.hardware': Topics related to IBM PC hardware.\n5. 'comp.sys.mac.hardware': Discussions about Mac hardware.\n6. 'comp.windows.x': Topics about the X Window System.\n7. 'misc.forsale': Posts related to buying and selling items.\n8. 'rec.autos': Discussions about automobiles.\n9. 'rec.motorcycles': Topics related to motorcycles.\n10. 'rec.sport.baseball': Discussions about baseball.\n11. 'rec.sport.hockey': Discussions about hockey.\n12. 'sci.crypt': Topics about cryptography and encryption.\n13. 'sci.electronics': Discussions about electronic systems and devices.\n14. 'sci.med': Topics related to medical science and healthcare.\n15. 'sci.space': Discussions about space and astronomy.\n16. 'soc.religion.christian': Topics about Christianity and related discussions.\n17. 'talk.politics.guns': Discussions about gun politics and related debates.\n18. 'talk.politics.mideast': Topics about politics in the Middle East.\n19. 'talk.politics.misc': General political discussions not covered by other categories.\n20. 'talk.religion.misc': Discussions about miscellaneous religious topics.\nFor the provided text below, determine the most appropriate category based on the descriptions above. Respond by selecting the label (e.g., alt.atheism, comp.graphics, etc.) that best matches the topic of the text. Provide the label in the first line and a brief explanation in the second line. The sentence: {sentence},

]

## C  TRAINING DYNAMICS AND CO-REGULARIZATION

**Training Dynamics**  The training dynamics during PLC fine-tuning (Stage I in Figure 1) is not only beneficial for clean and noisy sample separation (as we discuss in Section 3), but also contains rich information attributing to generative model learning (Stage II in Figure 1) (Zhuang et al., 2023). Leveraging such dynamics, our empirical objective becomes:

$$p(y|x) \propto \sum_{\hat{y}} p(\hat{y}|x)p(y|\hat{y}, W)$$

where $W$ denotes the training dynamics for each sample.

**Co-Regularization** Although we manage to mitigate the negative impact of label noises (Section 3,4), it is inevitable that small deviations in $p(\hat{y}|x)$ and $p(y|\hat{y}, x)$ could propagate to later stages, thus affecting the objective $p(y|x)$. We leverage multiple branches with identical architecture but different initializations (Zhuang et al., 2023). A co-regularization loss across branches is introduced to achieve consensus. Such a loss is calculated as the KL Divergence between the consensus probability (the average probability of models' predicted probability in different model branches) and each individual model's predicted probability. We apply co-regularization mechanism to both Stage I PLC $\mathbf{F}_{\varphi}(\hat{y}|x)$ and Stage II generative model $p_{\theta}(y|\hat{y}, x)$. To begin, we initialize $M$ copies of $\mathbf{F}_{\varphi}^{(m)}(\hat{y}|x)$ and $p_{\theta}^{(m)}(y|\hat{y}, x)$. Passing instances $x_i$ to different model branches, we can obtain corresponding model predicted probabilities $p_i^{(m)}$. Then, a aggregated probability $q_i$ can be calculated by averaging all predicted probabilities:

$$q_i = \frac{1}{M} \sum_{m=1}^{M} p_i^{(m)}$$

Given these, a co-regularization loss can be calculated as follows:

$$\ell_{\text{CR}} = \frac{1}{MN} \sum_{i=1}^{N} \sum_{m=1}^{M} \text{KLK}(q_i || p_i^{(m)})$$

$$= \frac{1}{MN} \sum_{i=1}^{N} \sum_{m=1}^{M} \sum_{c=1}^{C} q_{ic} \log \left( \frac{q_{ic} + \epsilon}{p_{ic}^{(m)} + \epsilon} \right)$$

where $\epsilon$ indicates a small positive number to avoid division by zero.

## D SiDyP Training Details

All experiments are conducted on CPU: Intel(R) Xeon(R) W-2295 CPU @ 3.00GHz and GPU: NVIDIA GeForce RTX A6000 GPUs using Python 3.11.5 and PyTorch 2.0.1. Table 6 indicates all specific hyper-parameters we use in different datasets. We use Adam (Kingma & Ba, 2017) as optimizer. $E_{\text{BERT}}$ is the training epochs for the BERT classifier. $E_{\text{SD}}$ is the training epochs for the simplex diffusion model. $\sigma$ is the estimated error rate in Algorithm 1. $\lambda$ is the threshold that we separate certain and uncertain prior in Algorithm 1. $\gamma$ is the threshold that we preserve the dominance candidates in uncertain prior in Algorithm 1. In Algorithm 2, $\alpha$ is the warmup epochs for Stage II generative model training. $m$ is the number of model branch. $\beta$ is the number of sample times that we use to refine our uncertain prior based on model's predictions.

**Time Complexity** We perform Big-O analysis for SiDyP. The time complexity for SiDyP is $O(W^2 \times T)$ where $W$ denotes the embedding size of training dynamics and $T$ is either training timesteps or inference timesteps of our simplex diffusion model. We choose $\gamma$ based on our empirical estimation. To make a fair comparison, we use the same estimate error rate in all other baselines which requires one. We grid search these hyper-parameters: $\lambda$ in [0.7, 0.8, 0.9, 1.0], $\gamma$ in [0.4, 0.6, 0.8], $\alpha$ in [1, 2, 3, 4, 5, 6], $\beta$ in [2, 4, 6, 8], $K$ in [10, 20, 30], train timesteps in [400, 500, 600, 700, 800], inference timesteps in [10, 20, 50, 100], learning rate in [1e-3, 6e-4, 3e-4, 1e-5].

| LLM ($\rightarrow$) | Llama-3-70b | | | | | | |
|---|---|---|---|---|---|---|---|
| Datasets ($\rightarrow$) | NumClaim | | TREC | | SemEval | | 20News |
| Method ($\downarrow$) | Zero-shot | Few-shot | Zero-shot | Few-shot | Zero-shot | Few-shot | Zero-shot |
| $E_{\text{BERT}}$ | 20 | 20 | 20 | 20 | 20 | 20 | 20 |
| batch size | 128 | 128 | 128 | 128 | 128 | 128 | 128 |
| learning rate (BERT) | 5e-5 | 5e-5 | 5e-5 | 5e-5 | 5e-5 | 5e-5 | 5e-5 |
| max length | 128 | 128 | 64 | 64 | 128 | 128 | 128 |
| $\sigma$ | 0.1 | 0.05 | 0.3 | 0.3 | 0.5 | 0.5 | 0.5 |
| $\lambda$ | 0.9 | 0.9 | 0.9 | 0.9 | 0.9 | 0.9 | 0.9 |
| $\gamma$ | 0.8 | 0.8 | 0.8 | 0.8 | 0.8 | 0.8 | 0.8 |
| $\alpha$ | 2 | 1 | 1 | 1 | 2 | 3 | 4 |
| $m$ | 3 | 3 | 3 | 3 | 3 | 3 | 3 |
| $\beta$ | 4 | 4 | 4 | 4 | 4 | 4 | 4 |
| $E_{\text{SD}}$ | 10 | 10 | 10 | 10 | 10 | 10 | 10 |
| batch size (SD) | 128 | 128 | 128 | 128 | 128 | 128 | 128 |
| learning rate (SD) | 6e-4 | 6e-4 | 6e-4 | 6e-4 | 6e-4 | 6e-4 | 6e-4 |
| train timesteps | 800 | 500 | 800 | 600 | 800 | 500 | 500 |
| inference timesteps | 10 | 10 | 50 | 80 | 10 | 10 | 10 |
| K | 20 | 20 | 20 | 10 | 10 | 10 | 10 |

Table 6: Training hyper-parameters details for SiDyP on all six Llama-3 generated datasets.

## E   LLM Noise Ratio

See Table 8

| LLM ($\rightarrow$) | Llama-3-70b | | | | | | |
|---|---|---|---|---|---|---|---|
| Datasets ($\rightarrow$) | NumClaim | | TREC | | SemEval | | 20News |
| Method ($\downarrow$) | Zero-shot | Few-shot | Zero-shot | Few-shot | Zero-shot | Few-shot | Zero-shot |
| Noise Ratio (Original) | 91.69 | 95.85 | 70.35 | 69.72 | 50.96 | 50.64 | 76.13 |
| No Answer Ratio | 0.00 | 0.00 | $3.6e^{-4}$ | $1.8e^{-4}$ | $2.5e^{-3}$ | $4.1e^{-3}$ | $1.4e^{-2}$ |
| Noise Ratio (After RA) | 91.69 | 95.85 | 70.35 | 69.72 | 50.96 | 50.64 | 76.23 |

Table 7: Llama-3-70b label noise ratio on training sets of NumClaim, TREC, and SemEval in zero-shot and few-shot manners, and 20News Group in zero-shot manner. "RA" represents random assignment.

| Dataset ($\rightarrow$) | SemEval | | | | | | | |
|---|---|---|---|---|---|---|---|---|
| Method ($\downarrow$) | Llama-3.1-70b | | Llama-3.1-405b | | GPT4o | | Mixtral-8x22b | |
| | Zero-shot | Few-shot | Zero-shot | Few-shot | Zero-shot | Few-shot | Zero-shot | Few-shot |
| Noise Ratio (Original) | 57.39 | 56.66 | 57.70 | 55.78 | 60.61 | 61.49 | 44.94 | 44.42 |
| No Answer Ratio | 0.00 | 0.00 | 0.001 | 0.0005 | 0.00 | 0.00 | 0.009 | 0.001 |
| Noise Ratio (After RA) | 57.39 | 56.66 | 57.75 | 55.78 | 60.61 | 61.49 | 44.94 | 44.42 |

Table 8: Label noise ratio of SemEval training set by Llama-3.1-70b, Llama-3.1-405b, GPT4o, and Mixtral-8×22b in both zero-shot and few-shot manners. "RA" represents random assignment.

## F   Label Candidate Efficacy

We calculate the accuracy of our label candidate compared to true labels for Llama-3-70b zero-shot labeled 20News Group, NumClaim, Trec, and SemEval across a wide-range of certain threshold $\lambda$ and dominant threshold $\gamma$. For certain candidate, the accuracy is easy to calculate as we can directly compare to its corresponding true label. For uncertain candidate, we either compare the

specific candidate with maximum probability with true label, or we check if true label lies in our uncertain candidate. Notably, when $\lambda = \gamma = 0$, our dynamic prior turns into fix prior. Our label candidate achieves an average of 9.5% improvement compared to fix prior. Figure 2 presents the entire distribution of our dynamic prior accuracy.

# G LLM-GENERATED LABEL NOISE CHARACTERISTICS

We plot SemEval's noise distribution of three different types of noise: LLM, synthetic, real-world in Figure 3. Except for real-world noise which has lower noise ratio (16%), both LLM-generated noise and synthetic noise's ratio are around 50%. Our observations are listed in the follow:

- Although the noise ratio of LLM-generated labels is comparable to that of synthetic noise, the correct ratio (the diagonal) is more diverse. In contrast, the correct ratios for all three types of synthetic noise are approximately 50%, reflecting an equal distribution of noise injection across classes.

- In synthetic noise, incorrect labels often show clear patterns (e.g., being consistently off by one class in ASN, noise distributed relatively equally in SN). The label noise introduced by IDN changes significantly depending on the seed used. Such a sensitivity to initial random state impacts model's robustness.

- While the distribution of synthetic noise indicates that this type of mislabeling often lacks contextual correlation, LLM-generated label noise reflects underlying relationships between classes (as evidenced by the similarity among the three LLMs), making it more aligned with real-world noise.

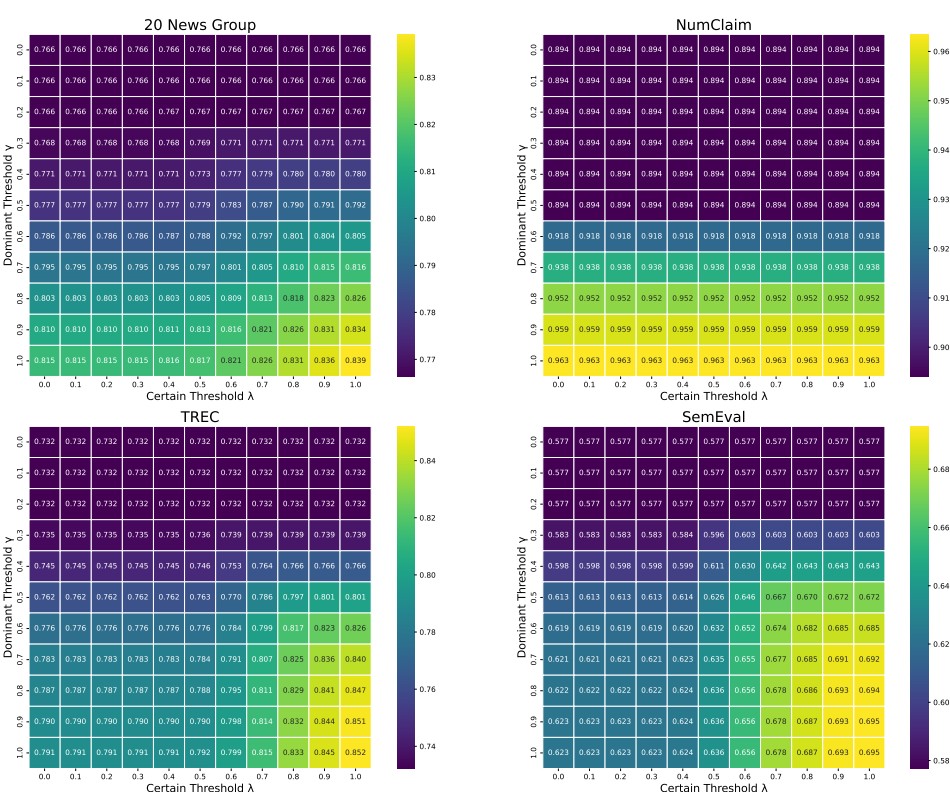

Figure 2: Label candidate accuracy distribution across different combinations of certain threshold $\lambda$ and dominant threshold $\gamma$ on 20 News Group, NumClaim, TREC, and SemEval labelled by Llama-3-70b in the zero-shot manner.

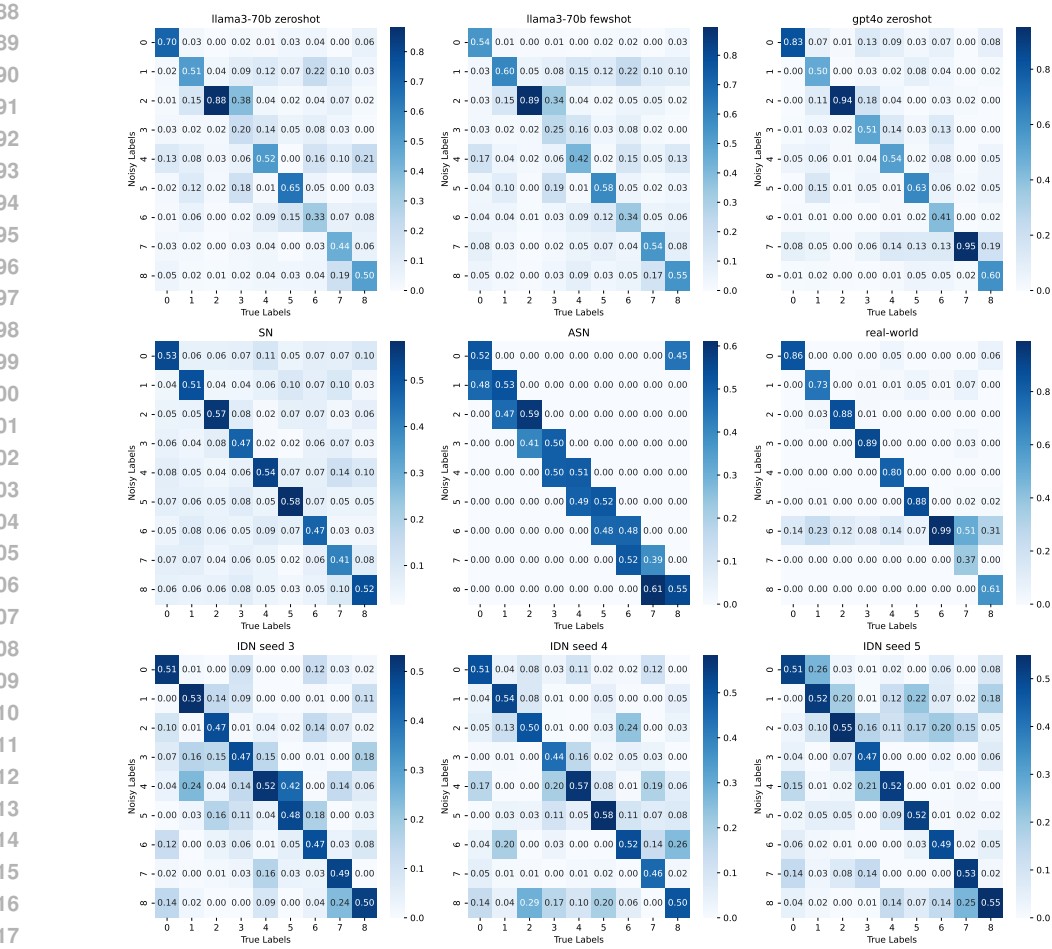

Figure 3: Noise distribution of differet types of noise: IDN under three seeds, Llama-3-70b zeroshot, Llama-3-70b fewshot, gpt4o, SN, ASN, and real-world

## H    CANDIDATE DISTILLATION EFFICACY

Figure 4 presents the performance increase brought by our candidate dynamic distillation algorithm. We use all four datasets labelled by Llama-3-70B. We obtain the amount of data instances in our training set of each dataset being corrected. The corrected uncertain ratio is calculated by such an amount dividing the total number of uncertain data instances which contains true labels in their candidates. We observe that more noise the datasete has, more significant improvement our distillation can bring. Notably, it is able to correct 8.6% label in SemEval few-shot prompting.

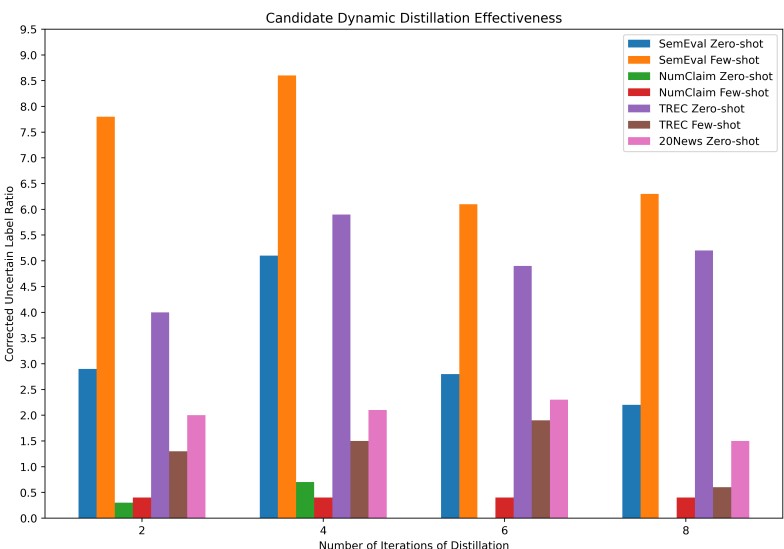

Figure 4: The ratio of uncertain labels being corrected by our candidate dynamic distillation.

