# OpenReview forum: "SiDyP: Simplex Diffusion with Dynamic Prior for Denoising Llama-Generated Labels"
_ICLR.cc/2025/Conference — Submitted to ICLR 2025_

### Official Review · Reviewer_a1Cc · 2024-11-02

**Soundness:** 3
**Presentation:** 2
**Contribution:** 2
**Rating:** 5
**Confidence:** 3

**Summary:**

This paper presents a work on learning with noisy labels in datasets annotated by LLMs. The proposed framework, SiDyP, is based on a noisy-supervised model called PLC and a simplex diffusion model. In this framework, the PLC model, fine-tuned with the noisy labeled data, is used to obtain dynamic training trajectories and induce a prior distribution, while the diffusion model serves as a posterior to iteratively calibrate PLC’s prior. Specifically, this paper introduces a true label candidates retrieval algorithm and a prior dynamic distillation
algorithm to mitigate the negative impact brought by noisy labels. Extensive experiments demonstrate that the proposed framework achieves competitive classification performance on multiple tasks.

**Strengths:**

1. Denoising LLM-generated labels may be a promising direction and is worth further investigation.
2. Experimental results are good, which shows that the proposed method is a state-of-the-art one.

**Weaknesses:**

1.The novelty of this paper is limited, capturing the label prior distribution based on training dynamics and using the simplex model for iterative refinement are the technical contributions of DyGen and TESS, respectively. The contributions of this paper seem like engineering innocations.

2.The proposed method introduces a few more hyperparameters. In practice, when we don't have a clean validation set, these hyperparameters can be difficult to set.

3.Writing needs to be improved. the dynamic prior needs to be clearly explained in the abstract and introduction, which can easily be confused with training dynamics. Additionally, the organization of Section 3 is not reasonable.

4.A more detailed ablation study would have been desirable, such as the number of candidate labels in Algorithm 1 and the number of iterations in Algorithm 2.

**Questions:**

1.Compared to DyGen, what are the main contributions of SiDyP? It seems that SiDyP just replaces the VAE part in DyGEN with a simplex diffusion model. There are no insightful innovations.

2.Line 165, why do we only consider the two candidate categories with the highest probabilities? In fact, the types and distributions of labels vary significantly across different tasks.

3.Some descriptions and settings of hyperparameters can be confusing. Does the estimated error rate refer to the noise ratio? What are the bases for setting these hyperparameters?

4.The symbols in Figure 1 are inconsistent with those in the main text.

5.Section 4.5, the effectiveness of the proposed label candidate retrieval algorithm has not been validated.

---

> ### Author Response · Authors · 2024-11-16
>
> Thank you for dedicating your time and effort to review our work. We appreciate your thoughtful feedback and recognition of our experimental results to be good and our proposed method to be state-of-the-art. We are excited that you believe the direction of our work is promising and open an avenue for further investigation. Below, we try to address your concerns, questions, and suggestions.
>
> **Limited Novelty**
> We would like to highlight the contribution of our framework:
> - We design two algorithms (label candidate retrieval and dynamic distillation) which introduce partial label learning into the field of learning-from-noisy-labels and validate their effectiveness in Section 5.5 in the updated draft.
> - We design a simplex denoising label diffusion model which converts noisy label into a reliable true label probability, allowing us to calibrate the predictions of fine-tuned classifiers over-fitted on noisy labels. We have also compared our simplex denoising label diffusion with a vanilla Gaussian diffusion model, which highlights the value of our framework. (See Section 4.5 and Table 5 in original submission or See 5.5 and Table 5 in the updated draft).
>
> **Hyperparameters finetuning difficulty**
> We are very glad that you point out this difficulty. Indeed, inspired by this paper [1], we are very aware that a clean validation dataset could affect the model selection, resulting in better performance in test dataset. In our work, the validation dataset that we use for model selection is also noisy. Therefore, our hyperparameters are tuned under a noisy validation dataset. We have incorporated this information in line 348-351 in our updated version.
>
> **Writing clarification**
> Thanks for your valuable feedback. We have clarified dynamic prior in our work to distinguish from training dynamics in line 21-25 in our updated version. We managed to re-organize Section 3 to make it more reasonable. (See Section 2, Section 3, Section 4, and Appendix C in our updated version).
>
> **Detailed ablation study**
> We would like to highlight that we have conducted ablation studies on our dynamic prior retrieval and distillation as well as our simplex diffusion models (See Table 5 in initial submission). We indeed demonstrate the efficacy of the number of candidates labels in Algorithm 1 in Appendix F (in initial submission). We appreciate your thoughtful suggestions of the number of iterations in Algorithm 2. We have incorporated it in Appendix H in the updated version.
>
> > 1. Compared to DyGen, what are the main contributions of SiDyP? ...
>
> Please see our response to your comments about novelty above (**Limited Novelty**)
>
> > 2. Line 165, why do we only consider the two candidate categories with the highest probabilities? ...
>
> Thank you for your careful reading of our paper. We do not **only** consider the two candidate categories with the highest probabilities. If the summation of the two highest probabilities is greater than $\gamma$, then we **eliminate** other candidates. Otherwise, we consider all the candidates. (See line 166-167 and line 11-17 in Algorithm 2 in initial submission). The purpose of preserving two candidates’ categories with highest probabilities when their summation is greater than $\gamma$ is to alleviate the uncertainty injected into generative model training. We also explored how $\gamma$ affects the efficacy of our label candidate retrieval in Appendix F (in initial submission).
>
> > 3. Some descriptions and settings of hyperparameters can be confusing. Does the estimated error rate refer to the noise ratio? What are the bases for setting these hyperparameters?
>
> Yes, the estimated error rate refers to the noise ratio. We have made the terminology consistent in our updated version (See Table 6 and line 1057-1058). We grid search the hyperparameters. We have incorporated the information of hyper-parameters grid search in line 1065-1070 in the update version.
>
> > The symbols in Figure 1 are inconsistent with those in the main text.
>
> We appreciate your careful reading of our paper. We have changed the symbols of the label in Stage II to be consistent with our section of simplex diffusion model (Section 4 in the updated draft). We would happy to address any other concerns you might have.
>
> > Section 4.5, the effectiveness of the proposed label candidate retrieval algorithm has not been validated.
>
> Thank you for the important comment. We have validated and discussed the efficacy of our label candidate retrieval algorithm in Section 5.5 (see Table5 and line 507-509) and Appendix F in the initial submission.
>
> [1] Zhu, Dawei, et al. "Weaker than you think: A critical look at weakly supervised learning." arXiv preprint arXiv:2305.17442 (2023).

---

### Official Review · Reviewer_uUcy · 2024-11-02

**Soundness:** 2
**Presentation:** 2
**Contribution:** 2
**Rating:** 5
**Confidence:** 3

**Summary:**

This paper proposes a method called SiDyP to address the learning-from-noisy-labels problem.
In the experiments, the authors validated the effectiveness of the proposed method using noisy data generated from the zero-shot and few-shot inference of Llama-3.

**Strengths:**

The authors conducted experiments using noisy labels generated by LLM Llama and demonstrated the effectiveness of the proposed method, which is one of the contributions of this paper.

**Weaknesses:**

1)	On the novelty of introducing LLMs for generating noisy data for learn-from-noisy-labels methods.
The authors claimed in the Introduction that, this is the first time that LLMs-generated datasets have been introduced in the realm of learning from noisy labels.
Actually, this statement is not rigorous.
There are already some works that focus on learning from noisy labels generated by LLMs [1,2,3].
These works concern the topic of Weak Supervision, which can be seen as a subset of the learning-from-noisy-labels problem; even though Weak Supervision considers the context of each one sample with multiple noisy labels, the learning-from-noisy-labels problem often refers to the context of one sample with one label.
Anyway, the authors should take these prior works into account while writing.

[1] Alfred: A System for Prompted Weak Supervision. ACL-2023 (demo).
[2] Combining prompt-based language models and weak supervision for labeling named entity recognition on legal documents. Artificial Intelligence and Law, 2024.
[3] https://snorkel.ai/blog/few-shot-learning-large-language-models/. Blog from Snorkel.


2) On the core research motivation and contributions.
Related to the first issue mentioned above, and more importantly, the core research motivation and contributions of this paper are not sufficiently clear:

    &emsp;2.1) If the designed method is more specifically tailored to noisy labels generated by LLMs (rather than noisy labels in the general learning-from-noisy-labels works), or in other words, if the proposed method is specifically designed for noisy labels generated by LLMs, then:
    The authors should analyze from theoretical or experimental perspectives whether noisy labels generated by LLMs have different characteristics compared to noisy labels in the traditional sense, and why the designed method is well-suited to handle these characteristics.

    &emsp;2.2) If the designed method is not specifically aimed at noisy labels generated by LLMs, then this paper essentially presents a new typical learning-from-noisy-labels method.

    &emsp;2.3) In summary, the authors do not clarify in the paper whether noisy labels generated by LLMs have different characteristics compared to traditional noisy labels, nor do they specify whether their contributions are specifically designed for noisy labels generated by LLMs.

**Questions:**

Please refer to the above Weaknesses.

**Details Of Ethics Concerns:**

None.

---

> ### Author Response · Authors · 2024-11-16
>
> We sincerely appreciate your time and effort in reviewing our work. We are happy that you recognize the effectiveness and contributions of our work addressing noisy labels generated by LLMs. We address your concerns below.
>
> **Lack of literature reviews of LLM annotations**
> Thank you for your insightful reviews. To clear any confusion, we would like to highlight that our work does not attempt to improve LLM’s annotation ability. Our broad research question is that given a noisy label dataset generated by LLM, how can we enhance the robustness of a classifier trained on this noisy dataset. Therefore, in the context of enhancing the robustness of a classifier training in a noisy dataset, we are the first to introduce the LLM-generated label noise to the best of our knowledge. We still greatly appreciate your feedback and have done a literature review of LLMs under weak supervision. We have incorporated them in line 57-58 and line 523-528 in the updated version and clarify our research question in line 57-67 and 136-140 in the updated version.
>
> **Unclear research motivation**
> We appreciate your constructive feedback. We would like to clarify that even though our study primarily focuses on LLM generated noise, our experiments show that it is applicable to any type of noise and outperforms all the baselines across all the noise types (synthetic, real-world, LLM). We have shown the empirical results in Table 4. Given that our approach is targeted to all kinds of noise, the reason why our work focuses on LLM-generated label noise because (1) Given LLM’s remarkable capabilities of in-context learning, there is an increasing potential to use it for initial data annotation instead of using other sources of weak labels. (2) It is a new type of noise that is under explored in the learning-from-noisy-label paradigm. We have clarified our research motivation (see in line 57-67 and 136-140) and contribution (see line 105-113) in the updated version.

---

### Official Review · Reviewer_sVro · 2024-11-03

**Soundness:** 2
**Presentation:** 3
**Contribution:** 2
**Rating:** 5
**Confidence:** 4

**Summary:**

The paper presents a novel approach using a simplex diffusion model for denoising labels generated by large language models (LLMs). The method aims to address the inherent noise in LLM-generated labels by leveraging an iterative denoising process within a continuous probability simplex space. The approach is motivated by the high cost and time demands of manual labeling, coupled with the practicality of LLMs for fast, albeit noisy, label generation. The paper introduces several key assumptions, including the local consistency of embedding spaces and the ability of diffusion models to gradually refine noisy labels into accurate representations. Experiments across different noise types, including symmetric, asymmetric, and instance-dependent noise, demonstrate the model's robustness. The results suggest that the method effectively reduces noise without requiring additional labeled data, making it a potentially valuable contribution to applications reliant on large-scale automatically generated labels.

**Strengths:**

The paper has several notable strengths across originality, quality, clarity, and significance:

Originality: The approach is innovative in applying a simplex diffusion model to iteratively denoise noisy LLM-generated labels. Using the probability simplex space to handle label noise is a creative application, enabling a gradual refinement of noisy labels, which is distinct from traditional denoising methods. This method also addresses the growing need for handling noise in labels generated by LLMs, making the approach highly relevant and timely.

Quality: The paper presents a structured approach with a combination of techniques, such as embedding-based candidate selection, to effectively manage noise. The experiment setup is comprehensive, covering multiple noise types (symmetric, asymmetric, instance-dependent), which reflects the robustness of the method. By comparing the proposed model to various baselines, the paper demonstrates a clear performance advantage in multiple scenarios, supporting the claim that the simplex diffusion model can enhance accuracy in noisy data environments.

Clarity: The overall structure of the paper is clear, with a logical flow from the problem formulation to the model design and experimental results. The motivation for the proposed method is well-presented, with a clear description of how each component of the simplex diffusion process contributes to the final denoised output.

Significance: The work is significant for applications that rely on LLM-generated labels, which often contain noise. By introducing a method that reduces noise without needing additional labeled data, the paper addresses a practical challenge, potentially improving the scalability and quality of labeling in real-world scenarios. This could be valuable across fields where large-scale data labeling is a bottleneck, such as natural language processing, computer vision, and medical imaging.

**Weaknesses:**

While the paper is promising, several areas could benefit from further improvement:

The paper lacks in-depth theoretical discussion regarding why the simplex diffusion model is particularly well-suited to the task of LLM label denoising. For example, the rationale behind using iterative denoising in the simplex space could be strengthened by exploring why this approach may perform better than existing denoising methods in managing specific types of noise common to LLM-generated labels.

Key Assumptions Verification: The paper is based on several assumptions, such as the local consistency of embeddings and the relationship between training dynamics and sample quality. However, these assumptions are not directly validated in the experiments. For instance, the assumption that noisy labels will have larger mean and standard deviation distances in the embedding space could be tested and quantified to provide stronger empirical support.

The paper could provide a more detailed analysis of the noise characteristics specific to LLM-generated labels. The motivation would be more convincing if it explained how these labels differ from other types of noise and why standard denoising methods are insufficient. Understanding these nuances would clarify why the simplex diffusion model is a more suitable choice.

Although the experiments include different noise types, they do not explore the method’s limitations in scenarios where noise is highly complex or includes extreme cases. Additionally, there is a lack of quantitative analysis regarding the training dynamics feature assumption, which would help validate the model’s robustness and provide further insights.

The paper lacks a failure analysis that would highlight the model's limitations. Discussing scenarios where the model underperforms could provide readers with a more realistic understanding of its applicability and suggest areas for future improvement.

**Questions:**

Noise Characteristics in LLM Labels: Could the authors provide more insights into the specific noise patterns in LLM-generated labels that this model targets? Understanding these patterns would clarify why traditional denoising methods might struggle with this type of noise.

Embedding Space Consistency Assumption: Can the authors validate the assumption that the local consistency of embeddings allows clean and noisy labels to be differentiated? This could be demonstrated by visualizing the embedding space to show the distribution differences between noisy and clean labels.

Training Dynamics Validation: Could the authors conduct a more detailed analysis of the training dynamics related to noisy samples? Quantifying the mean and standard deviation differences in distances for noisy versus clean samples would strengthen the paper’s empirical foundation.

Application Scenarios and Limitations: In which specific scenarios would the simplex diffusion model be most applicable or potentially limited? Additional examples would help readers understand the practical applicability of the method.

---

> ### Author Response · Authors · 2024-11-16
>
> We greatly appreciate your time and effort in reviewing our work. We are glad that you find our work of using probability simplex space to handle label noise to be innovative, and our experiments covering multiple noise types and various baselines to be comprehensive. We are happy that you recognize the significance and great potential of our work in real-world scenarios. Below, we try to address your concerns and questions.
>
> **Lack of theoretical discussion:**
> We would like to clarify that even though our study primarily focuses on LLM generated noise, our experiments show that it is applicable to any type of noise and outperforms all the baselines across all the noises (synthetic, real-world, LLM). We have shown such robustness in Table 4. We have discussed why our framework outperforms other methods in Section 4.5 (in initial submission; Section 5.5 in the updated version) by validating the effectiveness of both our dynamic priors and simplex diffusion. We have enriched it in Section 5.5 in the updated version.
>
> **Local consistency of embedding and training dynamics verification**
> Thank you for your comment. We would like to clarify that our work does not focus on introducing local consistency of embedding and training dynamics into learning-from-noisy-label problems. Local consistency of embedding is validated in the previous peer-reviewed papers ([1]) which we cite in our paper (see line 149-150). We also have validated this neighbor consistency by exploring our label candidate retrieval algorithm (See Appendix F). Training dynamics in our work is validated in [3].
>
> **LLM-generated label noise characteristics**
> We try our best to analyze how LLM-generated label noise is different from previously used synthetic noise and real-world noise. Please see Appendix G in the updated version. If you have any specific noise characterization in mind, we would be happy to incorporate it. We have enriched the discussion about why our framework works well in all these types of noise and the effectiveness of our simplex diffusion model in Section 5.5 in the update version.
>
> **Limitation analysis**
> We would like to highlight that our framework is tested for **four different datasets** with the number of classes ranging from 2 to 20, **five different LLMs** with labelling noise ratio from 5% to 60%, and **three different noise types** (synthetic, real-world, and LLM). We believe our experiments are comprehensive to explore our method’s capability under diversified scenarios. It would be more than appreciated if the reviewer could clarify and elaborate more about the highly complex noise and extreme cases for us to explore.
>
> **Failure analysis**
> We sincerely value your concern. We acknowledge that the improvement of our method is relatively marginal when the original noise ratio of the dataset is small.  For NumClaim dataset labelled by zero-shot Llama-3-70b with noise ratio of approximately 10%, SiDyP can only bring ~3% improvement compared to PLC. For SemEval dataset labelled by zero-shot Llama-3-70b with noise ratio of ~50%, SiDyP can enhance PLC performance by ~12%.
>
> > Noise Characteristics in LLM Labels: Could the authors provide more insights into the specific noise patterns in LLM-generated labels that this model targets? ...
>
> We have incorporated noise characteristics analysis in Appendix G in the updated version. Please refer to our response to **LLM-generated label noise characteristics** above.
>
> > Embedding Space Consistency Assumption: Can the authors validate the assumption that the local consistency of embeddings allows clean and noisy labels to be differentiated? ...
>
> We would like to point out that two assumptions in our paper are validated by these papers ([1]), which are cited in our initial submission. Please refer to our response to **Local consistency of embedding and training dynamics verification** above.
>
> > Training Dynamics Validation: Could the authors conduct a more detailed analysis of the training dynamics related to noisy samples? ...
>
> We would like to highlight that training dynmaics are validated by [2], which is cited in our initial submission. Please refer to our response to **Local consistency of embedding and training dynamics verification** above.
>
> > Application Scenarios and Limitations: In which specific scenarios would the simplex diffusion model be most applicable or potentially limited? ...
>
> We acknowledge that when the original noise ratio of the dataset is small, the improvement of SiDyP is relatively marginal. Please refer to our response in detail to **Failure analysis** above.
>
> [1] Ortego, Diego, et al. "Multi-objective interpolation training for robustness to label noise." Proceedings of the IEEE/CVF Conference on Computer Vision and Pattern Recognition. 2021.
>
> [2] Zhuang, Yuchen, et al. "Dygen: Learning from noisy labels via dynamics-enhanced generative modeling." Proceedings of the 29th ACM SIGKDD Conference on Knowledge Discovery and Data Mining. 2023.

---

### Official Review · Reviewer_7vxE · 2024-11-04

**Soundness:** 3
**Presentation:** 3
**Contribution:** 2
**Rating:** 5
**Confidence:** 3

**Summary:**

This paper proposes applying simplex diffusion to denoise annotations generated by large language models (LLMs). The core idea is to first obtain noisy annotations from LLMs and then iteratively refine these labels with a simplex diffusion model, which maps labels to a continuous probability space. The main experiments are conducted across four common text classification datasets using Llama 3.1-70b. Overall the results show that the proposed method achieves competitive performance in both zero-shot and few-shot settings. Additionally, the authors explore the method’s robustness by testing with different 4 LLM backbones and different types of label noises.

**Strengths:**

Overall, I think this paper is well-written and relatively easy to follow. The results are very competitive and clearly show the advantage of the proposed method. I think the construction of experiments is correct.  Additionally, the authors clearly communicate the problem and the potential impact of their approach on improving LLM-generated annotations, which would be a valuable contribution to the field.

**Weaknesses:**

= The comparisons with Mixtral models might not be entirely fair. If the authors used the base Mixtral-8x22b model, which is not instruction/chat-finetuned, it may not perform comparably to other models that have been finetuned on instruction data. For a fairer comparison, using Mixtral-8x22b-Instruct-v* or similar instruction-tuned models would better align with the capabilities of the other models tested.

= Insufficient literature coverage: Overall, this paper aims to solve the problem of LLM for data annotation and reduce annotation noise. I think it would be justifiable to have a better coverage of related work especially on LLM for data annotations. Give the limitation in space for the initial submission, I expect the authors to conduct a more rigorous literature survey of related work.

= Although the approach is novel for this specific problem of denoising llm annotaions, the method largely adapts simplex diffusion for signal denoising directly to noisy label refinement. This may raise questions regarding the technical novelty, as the adaptation itself does not appear to involve significant theoretical modifications.

**Questions:**

It would be helpful if the authors could clarify the following:

= For missing labels, why was a random label assignment chosen? Given the low failure rate, might it be preferable to disregard these cases instead?

= Were any specific system prompts or chat templates used when applying the prompts for LLM label generation? If so, could the authors provide details?

= In the few-shot experiments, the authors selected one example per label. Were these demonstration examples sampled from the training set or the validation set?

= For the 20News dataset, is there a reason why no results are reported for the few-shot experiments?

= Since the validation set is used for model selection, as an exploratory experiment, have the authors considered prompting the LLM with a larger few-shot sample (e.g., 50+ examples) from the validation set, given the support for long contexts in the LLMs used? It would be interesting to see how the efficiency of this approach compares with the proposed diffusion-based denoising pipeline.

---

> ### Author Response · Authors · 2024-11-16
>
> Thank you for dedicating your time and effort to review our work. We are grateful for your recognition of our competitive empirical results and correct experiment construction. We sincerely appreciate the positive comments on our paper being well-written and easy to follow. Below we try our best to address your concerns, questions, and suggestions.
>
> **Mixtral Model Comparison**
> We thank you for the careful review of our paper. We indeed use the instructed-finetuned Mixtral model (**Mixtral-8x22b-Instruct-v0.1**) in our experiment. We have provided the exact models that we use for each LLM in the updated version. See line 364-369 in the updated draft.
>
> **Insufficient Literature Coverage**
> We appreciate your insightful feedback. We have tried to add more related work on LLM data annotation and incorporated it in the updated version. We would be happy to add more of them if you have any specific ones you would like to suggest. See line 57-59 and 524-528 in the updated draft.
>
> **Technical Novelty Concern**
> We would like to highlight the contribution of our framework:
> - We design two algorithms (label candidate retrieval and dynamic distillation) which introduce partial label learning into the field of learning-from-noisy-labels and validate their effectiveness in Section 4.5 and Appendix F in the initial submission (Section 5.5, Appendix F, and Appendix H in the updated draft).
> - We design a simplex denoising label diffusion model which converts noisy label into a reliable true label probability, allowing us to calibrate the predictions of fine-tuned classifiers over-fitted on noisy labels. We have also compared our simplex denoising label diffusion with a vanilla Gaussian diffusion model, which highlights the value of our framework. (See Section 4.5 and Table 5 in original submission or See 5.5 and Table 5 in the updated draft).
>
> > For missing labels, why was a random label assignment chosen? Given the low failure rate, might it be preferable to disregard these cases instead?
>
> We choose random label assignments because we want to maintain dataset’s consistency. Although the failure rate is relatively low, different LLMs tend to have failure in different data instances. Leaving them out might result in benchmarking being done on different sets of subsamples of dataset. Moreover, we apply our framework to synthetic noise (See Table 4 in original submission) which does not leave out any data instances when injecting the noise. Therefore, we chose random label assignment instead of leaving those failure instances out to avoid the inconsistent comparisons caused by different dataset coverage under different noise types.
>
> > Were any specific system prompts or chat templates used when applying the prompts for LLM label generation? If so, could the authors provide details?
>
> We have added the chat templates in Appendix B in the updated draft. We provide one example below:
> ```
> prompt_json = [
> {"role": "user", "content": f“Classify the following sentence into 'INCLAIM', or 'OUTOFCLAIM' class. 'INCLAIM' refers to predictions or expectations about financial outcomes, it can be thought of as 'financial forecasts'. 'OUTOFCLAIM' refers to sentences that provide numerical information or established facts about past financial events. Now, for the following sentence provide the label in the first line and provide a short explanation in the second line. The sentence: {sentence}”},
> ]
> ```
> > In the few-shot experiments, the authors selected one example per label. Were these demonstration examples sampled from the training set or the validation set?
>
> The examples in a few-shot experiments are chosen from the training set.
>
> > For the 20News dataset, is there a reason why no results are reported for the few-shot experiments?
>
> Please refer to line 373-375 in initial submission (line 365-367 in updated draft) : “We only prompt 20News Group in zero-shot manner as it is a document level task, and Llama-3-70b has a context length limitation of 8192, which is not sufficient for few-shot learning.”
>
> > Since the validation set is used for model selection, as an exploratory experiment, have the authors considered prompting the LLM with a larger few-shot sample (e.g., 50+ examples) from the validation set, given the support for long contexts in the LLMs used? ...
>
> Thank you for your constructive suggestions. Inspired by this paper [1], we use a noisy validation set for model selection instead of a clean (gold) one. Therefore, comparing our model selected by a noisy validation set with LLMs with clean validation examples in few-shot prompting is less fair. We are grateful that you point out this and have clarified our choice of validation set in line 349-351 in the updated version.
>
> [1] Zhu, Dawei, et al. "Weaker than you think: A critical look at weakly supervised learning." arXiv preprint arXiv:2305.17442 (2023).

---

### Author Response · Authors · 2024-11-16
**Rebuttal by Authors**

We sincerely thank all the reviewers for their time, thoughtful comments, and constructive feedback. We are encouraged that the reviewers appreciate the following aspects of our work:

- Our work introduces a novel framework, SiDyP, which effectively addresses the challenge of noisy labels in datasets generated by large language models like Llama-3 (sVro).
- Our experimental setup is robust and comprehensive, providing a solid evaluation across various noise conditions and multiple text classification datasets. (a1Cc) The reviewers find that the experimental results are competitive and demonstrate state-of-the-art performance. (7vxE, sVro, uUcy, a1Cc)
- We have clearly communicated our contributions, and reviewers have found the paper well-written and easy to follow (7vxE, sVro).

We attempted our best to address the questions. We believe the comments and revisions have made the paper stronger and thank all the reviewers for their help. We try our best to highlight the revision we make in the rebuttal submission PDF. Please find individual responses to your questions below.

---

### Author Response · Authors · 2024-11-22
**Thank You and a Follow-Up to Reviewers**

Dear Reviewers,

Thank you for your positive reviews of our work.

We would appreciate your consideration of our responses to the comments you provided. We hope they address your concerns and aid in your overall assessment of our work.

Thank You,
Authors

---

### Author Response · Authors · 2024-12-04
**Rebuttal Summary**

Dear Reviewers, AC, SAC,

We sincerely thank the AC and SAC for their invaluable efforts in assigning reviews and guiding the process. We also deeply appreciate the reviewers for taking the time to read our paper and provide constructive feedback.

Although all reviewers remain silent during the discussion phase, here we summarize our rebuttal to present a high-level perspective that could hopefully help grasp our contribution and modification quickly.

- We focus on enhancing the learning from LLM-generated label noise, a new type of label noise that has not been explored before. Previous studies mainly focus on synthetic noise and real-world noise.
- We introduce label candidates and label distillation algorithm to improve the prior information ($y$) quality when estimating the posterior $p(y|\tilde{y}, x)$.
- We propose a simplex denoise label diffusion model to approximate the posterior $p(y|\tilde{y}, x)$, which provides a new perspective of reconstructing true labels from noisy labels.
- Extensive experiments across 4 NLP tasks, 5 LLMs, and 3 different types of noises are conducted to show the effectiveness of our framework. Although our focus is on LLM-generated label noise, we also test SiDyP under synthetic noise and real-world noise to demonstrate its robustness and applicability to all types of noises

We try our best to address all reviewers’ concerns and questions, and modify our draft accordingly:

- We add more related works on LLM data annotation and LLMs under weak supervision to address Reviewer uUcy and Reviewer 7vxE’s concern of insufficient literature coverage.
- We clarify our research question and motivation in Section 1&2 (Reviewer uUcy)
- We explain the term “dynamic prior” in our content and reorganize section 3 according to Reviewer a1Cc’s feedback on the presentation.
- We incorporate noise characteristic analysis of LLM-generated label noise to show its difference from other types of noise. (Reviewer sVro)
- We enrich our theoretical discussion in Section 5.5. (Reviewer sVro)
- We clarify our choice of using a noisy validation set for model selection, inspired by Zhu et al (2023) to address Reviewer a1Cc and 7vxE’s concerns.
- We expand the ablation experiments to show the effectiveness of our distillation algorithm (Reviewer a1Cc).
- We clarify our hyper-parameter search procedure. (Reviewer a1Cc).

Over the past month, we have put forth our best efforts to improve the quality of this paper and address each concern raised by the reviewers. We sincerely hope that our work will make a valuable contribution to the community. Thank you once again for your kind support and constructive feedback.

Sincerely,

The Authors



[1] Zhu, Dawei, et al. "Weaker than you think: A critical look at weakly supervised learning." arXiv preprint arXiv:2305.17442 (2023).

---

### Meta-Review · Area_Chair_Visx · 2024-12-22

**Metareview:**

This paper presents a new method for denoising labels generated by large language models (LLMs) using a simplex diffusion model. Reviewers mostly agreed that the paper is clearly written and the experiments are well designed. However, there are also some major limitations of this work, such as novelty, assumptions, theoretical analysis, missing related work, etc. Overall, the current version of this work is not ready for publication at ICLR.

**Additional Comments On Reviewer Discussion:**

Reviewers raised some major concerns on novelty, assumptions, theoretical analysis, missing related work, etc. The authors have provided detailed responses during the rebuttal and discussion period, which have addressed some of these concerns. However, some major concerns, such as novelty and theoretical analysis, still remain.

---

### Decision · Program_Chairs · 2025-01-22

Reject